# PuzzleWorld: A Benchmark for Multimodal, Open-Ended Reasoning in Puzzlehunts

**Hengzhi Li**[1,2] * **Justin Zhang**[1*] **Brendon Jiang**[1†] **Alexander Naehu**[1†] **Regan Song**[1†]
**Megan Tjandrasuwita**[1]   **Chanakya Ekbote**[1]   **Steven-Shine Chen**[1,2]
**Adithya Balachandran**[1]   **Wei Dai**[1]   **Rebecca Chang**[1]   **Paul Pu Liang**[1]
[1]MIT Media Lab and MIT EECS   [2]Imperial College London
`https://github.com/MIT-MI/PuzzleWorld`

## Abstract

Puzzlehunts are a genre of complex, multi-step puzzles lacking well-defined problem definitions. In contrast to conventional reasoning benchmarks consisting of tasks with clear instructions and constrained environments, puzzlehunts requires discovering the underlying problem structure from multimodal evidence and iterative reasoning, mirroring real-world domains such as scientific discovery, exploratory data analysis, or investigative problem-solving. Despite progress in foundation models, their performance on open-ended settings remains largely untested. We introduce PuzzleWorld, a comprehensive benchmark of 667 puzzlehunt-style problems designed to assess step-by-step, open-ended, and creative multimodal reasoning. Each puzzle is annotated with the final solution, detailed reasoning traces, and cognitive skill labels, enabling holistic benchmarking and fine-grained diagnostic analysis. Most state-of-the-art models achieve only 1-4% final answer accuracy. On PuzzleWorld, the best model solves only 18% of puzzles and reaches 40% stepwise accuracy, matching human puzzle novices but falling significantly behind puzzle enthusiasts. To demonstrate the value of our reasoning annotations, we show that fine-tuning a small model on reasoning traces boosts stepwise accuracy from 4% to 11%, which translates to improvements in downstream visual reasoning tasks. Our detailed error analysis reveals that current models exhibit myopic reasoning, are bottlenecked by the limitations of language-based inference, and lack sketching capabilities crucial for visual and spatial reasoning. We release PuzzleWorld at `https://github.com/MIT-MI/PuzzleWorld` to support future work on building more general, open-ended, and creative reasoning systems.

## 1 Introduction

Recent advances in language and multimodal reasoning (Liang et al., 2024b) have enabled significant progress in step-by-step problem-solving (Wei et al., 2022; Yao et al., 2023), transparent reasoning (Creswell & Shanahan, 2022; Luo et al., 2023), and enhanced human-AI collaboration (Wu et al., 2022; Chen et al., 2025b). Such progress has been fuelled by and evaluated on comprehensive benchmarks, particularly in domains like mathematics (Lu et al., 2024) and code (Jiang et al., 2024). However, these benchmarks are largely confined to narrow, well-defined environments. In coding, tasks are meticulously specified and validated within executable environments (Jimenez et al., 2024). In geometry, models often rely on domain-specific languages to structure their reasoning (Chervonyi et al., 2025). While valuable, these benchmarks primarily test a model's ability within a pre-defined problem space, rather than its ability to *discover the problem itself*.

In contrast, human reasoning excels in open-ended environments, where the rules are unstated and the objectives are ambiguous. We dynamically form hypotheses, adapt to implicit structures, and reason creatively across modalities to solve problems ranging from deciphering an escape room to novel scientific discovery. To build more generalist AI, we argue that the next frontier for evaluation lies beyond the current constrained settings. It demands benchmarks that challenge models to operate in less structured, discovery-driven environments that require more flexible and holistic reasoning (Mondorf & Plank, 2024).

---

*†These authors contributed equally.

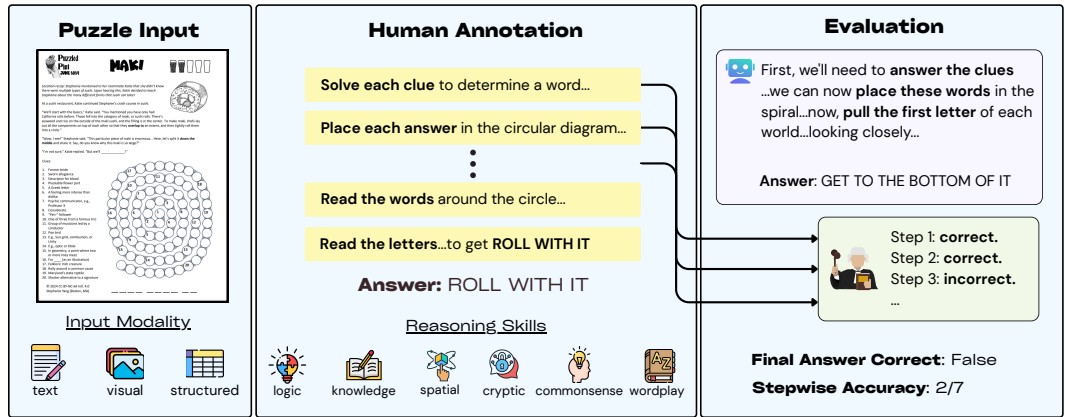

Figure 1: **Overview of PUZZLEWORLD**: PUZZLEWORLD is a dataset of complex puzzles that lack explicit instructions, requiring solvers to deduce the final answer from nuanced, multimodal cues from the puzzle content as well as external domain-specific knowledge. The raw puzzles and solutions are sourced from Puzzled Pint, and the solutions, which are PNG images, are transcribed into a sequence of reasoning steps by human annotators. These annotations enable us to measure the accuracy of the final answer and the step-by-step progress made towards the solution. Best-viewed zoomed in and in color, high-resolution puzzles are in Appendix E.3.

Puzzles are designed precisely to test these human-like generalist reasoning abilities. While some popular puzzles such as Sudokus and crosswords are rigidly formatted, others, like *puzzlehunts*, are intentionally open-ended. In a puzzlehunt, puzzle solvers are not given a clear task; they must first infer the nature of the problem from ambiguous clues embedded in text, images, or cultural references before devising and executing a solution.

Beyond their entertainment value, puzzlehunts model the essential challenges of real-world discovery and analysis. They demand compositional thinking, lateral reasoning, and the resilience to pursue leads, backtrack from dead ends, and manage uncertainty. Unlike current AI benchmarks that present well-specified tasks, puzzlehunts compel solvers to discover both *what* the problem is and *how* to solve it. This dual challenge makes them uniquely suited for evaluating general-purpose reasoning systems under conditions that more closely resemble open-ended scenarios like scientific investigation, intelligence analysis, or exploratory design.

To bridge this gap, we introduce PUZZLEWORLD, a benchmark of 667 real-world puzzlehunt problems curated from Puzzled Pint (Puzzled Pint, 2025), a monthly puzzlehunt event with content released under a Creative Commons license. These puzzles offer an open-ended, compositional challenge beyond prior benchmarks focused on instruction-following or task completion, and will grow with new puzzle releases. For each puzzle, we provide fine-grained annotation of its solution, input modalities, cognitive reasoning skills it exercises, and a manually curated step-by-step solution trace. These rich annotations support diagnostic analysis, model training, and detailed evaluation of models' reasoning capabilities. An overview of PUZZLEWORLD is provided in Figure 1.

PUZZLEWORLD enables us to systematically study the multimodal and multi-step reasoning capabilities of today's best foundation models. Most state-of-the-art models achieve only 1-4% final answer accuracy, with the best model solving only 14% of puzzles and reaching 40% stepwise accuracy. We additionally find that detailed annotations are important, as fine-tuning a model on annotated reasoning traces significantly improves a small model's performance, both within PUZZLEWORLD and on other visual reasoning datasets. We also conduct detailed error analysis on models' performance on PUZZLEWORLD, yielding tangible directions for future work in improving multimodal open-ended reasoning in AI. Together, these elements position PUZZLEWORLD as a rigorous resource for evaluating and improving general-purpose multimodal reasoning in AI systems. In the long run, we believe PUZZLEWORLD can catalyze more general and adaptable AI for mathematical and logical reasoning, open-ended scientific discovery, and assistive agents. PUZZLEWORLD is publicly available at https://github.com/MIT-MI/PuzzleWorld.

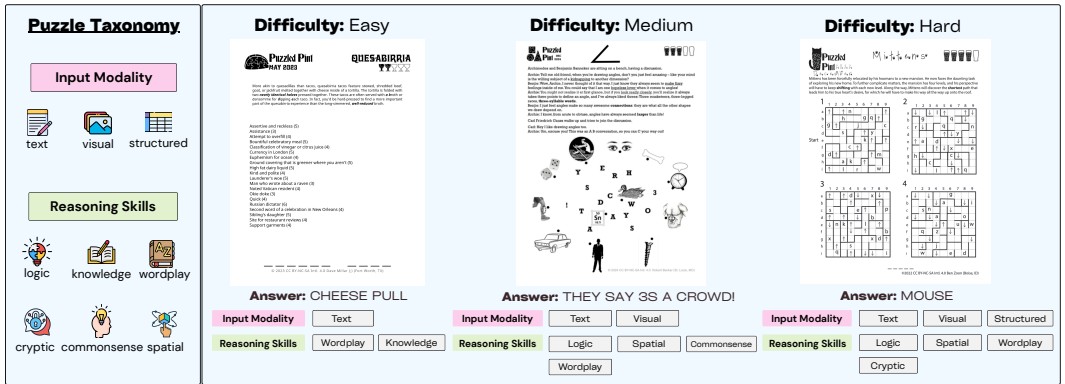

Figure 2: **Overview of samples from PUZZLEWORLD. Left:** To gain a deeper understanding of model performance on PUZZLEWORLD, each puzzle is annotated with the input modalities of the puzzle content, the reasoning skills required to solve the puzzle, and step-by-step reasoning steps. **Right:** Example modality and reasoning skill annotations on three puzzles. High-resolution puzzle images are included in Appendix E.3.

## 2 RELATED WORK

**Large Language Model (LLM) Reasoning.** LLMs have demonstrated remarkable emergent capabilities, often matching or even surpassing human performance across a wide range of tasks (Street et al., 2024). Notably, models such as GPT-4 (Achiam et al., 2023) and Claude (Anthropic, 2025) have achieved strong results not only on traditional NLP benchmarks—like question answering, summarization, and translation (Widyassari et al., 2022; Soares & Parreiras, 2020; Singh et al., 2017), but also in more complex domains such as mathematical reasoning, programming, and logical deduction (Ahn et al., 2024; Jiang et al., 2024; Lam et al., 2024). These abilities suggest that LLMs are beginning to exhibit general-purpose reasoning skills, making them increasingly relevant to both academic research and practical applications. However, despite these impressive capabilities, understanding the full extent and limitations of LLM reasoning remains a crucial open question, underscoring the need for benchmarks that rigorously assess their capability for flexible, holistic reasoning (Mondorf & Plank, 2024; Chang et al., 2024).

**Reasoning Benchmarks.** Numerous reasoning benchmarks have been proposed to evaluate various cognitive skills, including visual mathematical reasoning (Lu et al., 2024), spatial understanding (Wang et al., 2024a), analogical reasoning (Yiu et al., 2024), and social reasoning (Li et al., 2025; Mathur et al., 2025). However, few have addressed abstract, open-ended problems that demand holistic reasoning. HEMM (Liang et al., 2024a), SciBench (Wang et al., 2024b), MMMU (Yue et al., 2024a), MMMU-Pro (Yue et al., 2024b), MMT-Bench (Ying et al., 2024), and Olympiad-Bench (He et al., 2024) test multimodal reasoning across various disciplines in academic and real-world contexts. While these tasks are broad and challenging, they typically involve well-defined questions that closely resemble the training distributions of large models. As such, they primarily assess in-distribution reasoning rather than creativity or adaptability. ARC-AGI (Chollet, 2019) tests the ability to reason and adapt to new situations through abstract visual pattern recognition tasks that require minimal prior knowledge, yet it lacks the open-ended, exploratory nature of real-world problem solving. In contrast, PUZZLEWORLD targets open-ended reasoning through puzzlehunts that lack explicit instructions. Solving these tasks requires creatively piecing together subtle hints, often across many modalities, into coherent multi-step reasoning chains.

**Puzzle Benchmarks.** A growing line of work has explored the use of puzzles to test the reasoning capabilities of AI systems. PuzzleVQA (Chia et al., 2024) consists of 2,000 puzzles that require abstracting patterns from visual puzzles to answer multiple-choice questions. AlgoVQA (Ghosal et al., 2024) is a visual puzzle benchmark requiring algorithmic reasoning. PUZZLES (Estermann et al., 2024) tests the ability of RL agents to perform algorithmic reasoning on a set of 40 puzzles. While valuable for evaluating specific skills, these benchmarks focus on narrow domains with constrained task formats, and modern models generally perform well on these benchmarks (Chia et al., 2024; Moskvichev et al., 2023; Yue et al., 2024a). On the other hand, the unstructured nature of the puzzlehunt problems in PUZZLEWORLD requires models to interpret ambiguous cues, explore

| Statistic | Value |
|---|---|
| Total # of puzzles | 667 |
| Avg. # of Reasoning Steps | 5.4 |
| Percent # of Visual Reasoning Steps | 12.3% |
| Avg. Word Count per Reasoning Step | 22.5 |
| Correlation between Difficulty and # of Reasoning Steps | 0.24 |

Figure 3: **Dataset construction procedure and statistics: Left:** First, we source raw puzzles and solutions from Puzzled Pint. As the Puzzled Pint solutions are often not correctly parsed by OCR, each puzzle's metadata and reasoning steps are human-annotated. We use GPT-4o to automatically flag ambiguous and inconsistent annotations. Finally, two human verifiers perform a manual data cleaning on the flagged puzzles to ensure a consistent annotation format. **Right:** We summarize the statistics of our dataset. The average number of reasoning steps is high, and the steps are relatively complex, as shown by the high average word count.

creative strategies, and integrate information across diverse modalities and knowledge areas. In contrast to previous benchmarks that isolate strictly structured vertical reasoning (Chen et al., 2025a) or narrative-based lateral thinking (Huang et al., 2024), puzzlehunt puzzles require an integrated blend of lateral thinking, symbolic abstraction, and visual–spatial reasoning. The closest to our benchmark is EnigmaEval (Wang et al., 2025), which also evaluates AI's reasoning capabilities on puzzlehunts. However, EnigmaEval is a closed-source evaluation-only dataset and does not include manually annotated step-by-step solutions. The open-sourced puzzles and rich annotations in PUZZLEWORLD support fine-grained analysis of intermediate reasoning and failure modes, facilitating the development and evaluation of more robust, general-purpose reasoning models.

## 3 TAXONOMIZING MULTIMODAL REASONING IN PUZZLEHUNTS

To understand how solving puzzlehunts engages reasoning capabilities evaluated separately in benchmarks like MMMU (Yue et al., 2024b), we analyze puzzle solutions and classify them along two dimensions: input modality and reasoning mechanism. This taxonomy provides a comprehensive evaluation framework that captures both the form in which information is presented and the cognitive strategies required for reasoning.

### 3.1 PUZZLE INPUT MODALITIES

We consider three puzzle input modalities: **Text**, encompassing textual information such as instructions, narratives, or word puzzles, testing the model's ability to extract relevant linguistic information; **Visual**, which includes unstructured visuals like images, icons, and typography, challenging the models to interpret visual semantics and patterns; and **Structured**, which refers to systematically organized visual information, such as tables, graphs, grids, matrices, and charts. Table 1 shows the distribution of puzzles across modality and difficulty.

### 3.2 PUZZLE REASONING MECHANISMS

We identify six core cognitive abilities essential for effective puzzle-solving in PUZZLE-WORLD. These include **logic**, which covers inferential reasoning such as deduction and causal inference; **wordplay**, involving flexible linguistic interpretation through puns, anagrams, and homophones; **spatial reasoning**, which tests an AI's ability to mentally manip-

Table 1: **Count of puzzles across modalities and difficulties.** Across all modalities, the distribution of difficulties is similar.

| | Easy | Medium | Hard |
|---|---|---|---|
| Text | 131 | 322 | 151 |
| Visual | 90 | 226 | 111 |
| Structured | 59 | 181 | 108 |

ulate objects and navigate structures; and **cryptic decoding**, which requires recognizing and applying transformations like ciphers and hidden encodings. In addition, **knowledge-based reasoning** leverages domain-specific facts from areas such as science or history, while **commonsense reasoning** draws on implicit real-world expectations. This taxonomic approach enables targeted evaluation and analysis of AI reasoning capabilities across different cognitive dimensions. By mapping specific puzzles and reasoning tasks to combinations of modalities and mechanisms, we can identify areas of strength and weakness in AI systems, track progress over time, and guide future development efforts toward more balanced reasoning capabilities.

## 4 CREATING PUZZLEWORLD

### 4.1 DATA COLLECTION AND PRE-PROCESSING

We collected our puzzle corpus from Puzzled Pint (2025), an organization that publishes puzzles under Creative Commons (CC BY-NC-SA Intl. 4.0). Their repository contains monthly puzzles designed for collaborative solving, covering a diverse range of puzzle types and difficulties. This allowed us to obtain more than 700 raw puzzles spanning from 2010 to 2025.

Each puzzle in our dataset consists of its original PDF containing the puzzle content, a single-phrase answer, and a solution document. Unlike Wang et al. (2025), we deliberately preserved the original puzzle format rather than transcribing content into separate text and images. This decision was motivated by the importance of spatial relationships in puzzle layouts to the solving process. Furthermore, Wang et al. (2025) showed that the best foundation models are not primarily constrained by OCR capabilities. Instead, we devote our manual effort to construct fine-grained annotations of puzzle reasoning steps, ensuring that the annotations accurately capture the intended solution pathways while maintaining the integrity of the original puzzle presentation.

### 4.2 DATA ANNOTATION

To facilitate AI's reasoning capabilities, we designed a comprehensive annotation structure for PUZZLEWORLD. Each puzzle is represented by a standardized metadata and visual assets. To prevent ambiguity, we discard puzzles that have incomplete solutions, multiple ground truth answers, or require physical activity to solve the puzzles. This leaves us with 667 annotated puzzles.

#### 4.2.1 METADATA SCHEMA

Each puzzle is annotated using a JSON schema comprising several fields: a descriptive **title**; **flavor text** providing narrative context; a **difficulty** label (easy, medium, or hard); **solution** representing the canonical answer; a **reasoning** field of an ordered sequence of steps leading to the solution; a **modality** tag specifying the input types involved; a list of **skills** capturing the cognitive abilities required for solving; and a **source** field attributing the data origin. Figure 4 illustrates an example annotation.

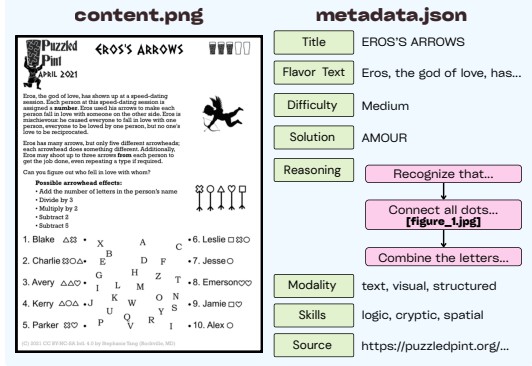

#### 4.2.2 REASONING ANNOTATION

A key contribution of our annotations is the decomposition of puzzle-solving into reasoning steps. Each step is formalized as a tuple $\langle e, f \rangle$ where $e$ represents the text explanation and $f$ denotes an optional figure illustrating the reasoning. To ensure consistency, we loosely require each step to begin with an atomic operation, such as pattern discovery or sketching, followed by the operation's intermediate output. This annotation structure enables fine-grained reasoning trajectory analysis on AI solutions.

Figure 4: **Illustration of metadata schema:** All puzzles are annotated with accompanying metadata, which includes the title, flavor text, difficulty, final answer, reasoning steps, input modalities, reasoning skills, and the link to the puzzle.

### 4.3 VERIFICATION OF ANNOTATIONS AND DATA CONTAMINATION

To ensure annotation quality and integrity, we implemented a two-stage verification protocol. First, we used GPT-4o to flag each puzzle annotation for correctness and reasoning coherence. This automated screening identified reasoning steps exhibiting ambiguity or logical discontinuities that might impede systematic analysis, which has flagged 12.11% of the dataset. Subsequently, two human verifiers independently reviewed all flagged annotations, applying corrections where necessary. This verification process resulted in modifications to 10.93% of the initially annotated puzzles. As an additional quality assurance measure, we conducted manual verification of a random subset comprising 5% of the dataset. In this evaluation, 96.5% of the verified annotations are marked as correct by the verifiers, demonstrating the high reliability of our annotation methodology. Finally, we verify whether frontier models has memorized any of the puzzles in PUZZLEWORLD. We describe our procedure in E.1, where we find no evidence of data contamination.

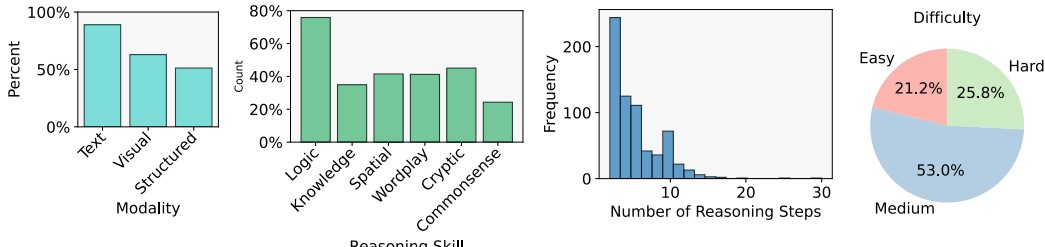

Figure 5: **PUZZLEWORLD dataset statistics.** Distributions of modalities and reasoning skills are balanced. While the majority of puzzles are of medium difficulty, there is significant number of easy and hard puzzles. The number of reasoning steps follows a long-tail distribution, with many solutions requiring more than 5 steps and some hard puzzles requiring up to 30 steps of reasoning.

## 4.4 DATASET STATISTICS

We summarize key statistics in Figure 3 (right). The average number of reasoning steps is above 5, and the average word count per reasoning step is above 20, demonstrating the complexity of the reasoning traces. Additionally, 12.3% of the steps have a visual intermediate output, highlighting the importance of sketching and spatial reasoning to solve puzzles. The correlation between puzzle difficulty and number of reasoning steps is 0.24. While we expect difficulty and number of reasoning steps to be positively correlated, the magnitude of the correlation is relatively low, as the difficulty of the puzzles also stems from their open-ended nature. Figure 5 shows the distribution of puzzles by modalities, reasoning skills, number of reasoning steps, and difficulty.

## 5 EXPERIMENTS

In this section, we evaluate frontier closed and open-source multimodal LLMs on the PUZZLE-WORLD dataset. We detail the evaluation setup, present quantitative results, and conduct qualitative error analysis to understand model behavior in open-ended, multimodal puzzle reasoning.

## 5.1 EXPERIMENTAL SETUP

We evaluate frontier closed-source reasoning models on PUZZLEWORLD, including GPT-o3 (OpenAI, 2025), GPT-4o (Achiam et al., 2023), Claude Opus 4 (Anthropic, 2025), Gemini-2.5-Pro (Comanici et al., 2025), Gemini-3-Pro (Google, 2025), and Grok 4 (xAI, 2025). We also evaluate open-source models Qwen QVQ (Qwen, 2024), InternVL3 (Zhu et al., 2025), and Kimi VL A3B (Team et al., 2025). We prompt each model with a comprehensive prompt as in Wang et al. (2025), followed by the puzzle images and transcribed flavor text. See Appendix F for the evaluation prompt.

We also provide a human baseline on PUZZLEWORLD, considering three tiers of puzzlehunter expertise: **Novice**, with no prior puzzlehunt experience; **Enthusiasts**, who showed interest or have occasionally participated (1-2 sessions) in puzzlehunts, and **Experts** among the top teams at monthly Puzzled Pint meetings. We we gathered 9 Novices and 9 Enthusiasts across high school and college ages. We sampled 5% puzzles from PUZZLEWORLDand assigned each participant to solve four puzzles. Participants were given an hour to solve each puzzle, matching the usual expected time at a live session, and were asked to provide paragraph explanations for their solution. For Experts, we use statistics from Puzzled Pint sessions in Syracuse, New York, and Bangalore, India, dating from January 2023 to June 2025. The statistics suggest that expert puzzlehunters consistently solve all five puzzles within one to two hours, which is on average less than the time prescribed to our human participants. We thus assume that human Experts achieve perfect accuracy on PUZZLEWORLD.

### 5.1.1 AUTOMATIC EVALUATION METRICS

Beyond final answer accuracy, we additionally evaluate the models' *stepwise accuracy* by comparing their solution with the annotated ground truth reasoning steps. Since puzzles can have multiple solution pathways, we define the stepwise accuracy score of a candidate solution to be the *last* annotated reasoning step it successfully executed out of all the reasoning steps. We implement an LLM judge (Zheng et al., 2023) with GPT-4o to determine the stepwise score of each candidate solution. For each reasoning step in the reference solution, the LLM judge determines if the step is met by the candidate response. To evaluate LLM judge's reliability, we compared its stepwise evaluations on 20 random puzzles against human evaluations. The LLM judge achieved a Pearson correlation of $r = 0.829$ ($p = 6.3 \times 10^{-6}$) and a mean absolute error (MAE) of $0.083$ with respect to human scores, indicating strong alignment with human judgment.

Table 2: **Model performance.** Accuracy (Acc) and stepwise accuracy (Step) are reported overall and per modality. Models struggle significantly on PUZZLEWORLD, most achieve only 1-4% answer accuracy, with closed-source models generally outperforming open-source ones. The best model, Gemini 3 Pro, solves only 18% of puzzles and reaches 40% stepwise accuracy, matching human Novice performance but significantly falling behind Enthusiasts.

|  | | Overall | | Text | | Visual | | Structured | |
|---|---|---|---|---|---|---|---|---|---|
|  | Model | Acc | Step | Acc | Step | Acc | Step | Acc | Step |
| *Open* | QVQ-72B-Preview | **1.36** | **30.23** | **1.33** | **29.25** | 0.63 | **27.96** | 1.18 | **32.40** |
|  | InternVL3-78B | 0.89 | 15.49 | 0.83 | 14.80 | 0.47 | 14.48 | 1.15 | 17.97 |
|  | Kimi VL A3B | 1.33 | 19.10 | 1.16 | 17.91 | **0.94** | 18.84 | **1.72** | 21.41 |
| *Closed* | GPT-o3 | 14.22 | 39.81 | 15.16 | **39.92** | 8.96 | 33.38 | 13.53 | **41.28** |
|  | GPT-4o | 1.83 | 22.09 | 1.92 | 20.00 | 0.73 | 20.20 | 2.77 | 28.09 |
|  | Claude Opus 4 | 4.50 | 24.56 | 4.20 | 23.77 | 4.04 | 22.60 | 4.37 | 26.93 |
|  | Gemini 2.5 Pro | 7.65 | 31.61 | 8.07 | 31.09 | 4.99 | 29.06 | 6.71 | 32.34 |
|  | Gemini 3 Pro | **18.00** | **39.99** | **20.30** | 39.34 | **14.71** | **38.81** | **20.25** | 39.99 |
|  | Grok 4 | 3.33 | 13.79 | 3.85 | 13.64 | 3.70 | 14.19 | 1.56 | 11.22 |
| *Human* | Human Novice | 13.89 | 23.10 | 16.98 | 25.32 | 11.00 | 22.70 | 16.67 | 24.92 |
|  | Human Enthusiast | 44.44 | 51.70 | 44.14 | 52.58 | 44.00 | 52.20 | 54.17 | 57.81 |
|  | Human Expert | **100.0** | **100.0** | **100.0** | **100.0** | **100.0** | **100.0** | **100.0** | **100.0** |

## 5.2 RESULTS

### 5.2.1 OVERALL PERFORMANCE OF FRONTIER MODELS

We report the models' performance in Table 2. All models exhibit extremely low final answer accuracy on PUZZLEWORLD, with most achieving close to 1-4%. Gemini 3 Pro attains the highest overall accuracy at 18.00%, matching human Novice performance, while the best-performing open-source model, QVQ-72B-Preview, reaches just 1.36%. All models perform significantly worse than human Enthusiasts and Experts. Although the uniformly low accuracies underscore our benchmark's difficulty, it offers limited insight into the models' reasoning capabilities.

To address this, our stepwise evaluation metrics provide a more nuanced view of models' reasoning performance. These metrics reveal that models with poor final answer accuracy, such as InternVL3, still demonstrate good intermediate reasoning, achieving up to 15.49% stepwise accuracy. Similarly, while QVQ-72B-Preview lags behind all closed-source models in final answer accuracy, it outperforms many of them in stepwise accuracy (30.2%), reflecting coherent and promising reasoning capabilities despite not reaching the correct final output. The combination of final answer and stepwise metrics enable PUZZLEWORLD to remain highly challenging while offering detailed diagnostics for model evaluation and development.

In terms of input modalities, models generally perform best on text-based puzzles, with significantly lower accuracy on puzzles involving unstructured visual inputs. Interestingly, most models achieve better performance on structured puzzles, such as crosswords where the spatial format constrained, over unstructured visual puzzles. In contrast, puzzles involving free-form visuals remain difficult, with models often achieving less than half their text puzzle accuracy on these inputs. Additionally, in Appendix D.2, we find that these modality-level performance discrepancies are not systematically driven by optical character recognition (OCR) limitations. These trends highlight current models' persistent weaknesses in visual grounding and spatial reasoning. In contrast, we observe that visual and structured modalities do not pose additional difficulty for human solvers regardless of their puzzle solving experience.

We further analyze model performance by puzzle difficulty in Appendix E.2. We find that, as expected, model accuracy decreases as difficulty increases. Interestingly, GPT-o3 performs best on easy, structured puzzles – likely because of their clean, regular layouts – whereas even harder structured puzzles remain challenging as they become more irregular and visually complex.

## 5.3 IMPROVING REASONING ON DOWNSTREAM TASKS WITH PUZZLEWORLD

To explore whether PUZZLEWORLD can support improvement in model reasoning, we fine-tuned an 8B InternVL3 model with supervised fine-tuning on annotated reasoning traces from 80% of the dataset, and evaluated performance on the 20% held-out test set. As a control, we fine-tuned the same model using only the final answers, without access to reasoning traces. Full training details are provided in Appendix F.2.

Our results in Table 3 highlight the value of PUZZLEWORLD's stepwise annotations. Fine-tuning on reasoning traces doubles the model's stepwise accuracy – from the base model's 4.78% to 11.00%. In contrast, fine-tuning on final answers alone impairs performance, reducing stepwise accuracy to 2.96% and driving answer accuracy to zero. Despite the improved stepwise accuracy, the fine-tuned model's answer accuracy remained at 0.76%. This suggests that while fine-tuning enhanced model's intermediate reasoning, it was insufficient to solve additional puzzles completely. This result underscores both the difficulty of PUZZLEWORLD and the limitations of straightforward fine-tuning approaches in addressing such complex reasoning challenges in our dataset.

Table 3: **Fine-tuning on PUZZLEWORLD.** Fine-tuning InternVL3 on PUZZLEWORLD reasoning traces significantly improves its stepwise accuracy, while fine-tuning on final answer only causes model reasoning to collapse.

| Model | Acc. | Step. |
|---|---|---|
| Base | **0.76%** | 4.78% |
| Fine-tuned (Answer-only) | 0.00% | 2.96% |
| Fine-tuned (Reasoning) | **0.76%** | **11.00%** |

Table 4: **PUZZLEWORLD fine-tuned model performance on downstream reasoning tasks.** Fine-tuning InternVL3 on PUZZLEWORLD leads to performance gains on visually-oriented tasks, such as Rebus puzzles, geometry, and visual question answering, but slightly reducing performance on problems more dependent on external knowledge, such as textbook question answering.

| Dataset | Task | Base Model | Fine-tuned on PUZZLEWORLD |
|---|---|---|---|
| Rebus Puzzles | Puzzle reasoning | 3.2% | **5.1%** |
| MathVista | Geometry problem solving | 65.87% | **66.35%** |
| | Textbook question answering | **63.92%** | 60.13% |
| | Math word problem | **62.37%** | 59.14% |
| | Visual question answering | 32.40% | **39.11%** |

We then explore whether PUZZLEWORLD's detailed stepwise annotation can improve models on downstream reasoning tasks. We finetuned a model on 80% of PUZZLEWORLD and evaluated it on two benchmarks: a Rebus puzzles dataset (Lee et al., 2025) involving visual metaphors without explicit instructions, and the MathVista dataset (Lu et al., 2024), ranging from general visual question answering to domain-specific geometry questions. Our results are shown in Table 4.

Finetuning on PUZZLEWORLD yields notable performance gains on Rebus puzzles, where the model's accuracy increased from 3.2% to 5.1%. On MathVista, the fine-tuned model shows significant improvement in geometry problem solving and visual question answering, but its performance slightly decreased on tasks that are outside of PUZZLEWORLD's reasoning skills and require external knowledge, such as textbook question answering and math word problems. This performance improvement suggests that the skills learned from PUZZLEWORLD are not merely task-specific. They represent transferable, general-purpose reasoning capabilities, making our dataset a reflection of diverse reasoning expertises and a valuable tool for enhancing models' capabilities.

## 5.4 DETAILED ERROR ANALYSIS

In the following section, we highlight the main sources of errors by frontier reasoning multimodal LLMs on PUZZLEWORLD: myopic reasoning, limitations of language, and lack of visual sketching capabilities. We focus our analysis on GPT-o3, and we provide qualitative examples for each error category in Figure 6. In Appendix D.2, we provide a less common "visual forgetting" error pattern, where model deviates from visual puzzle content during extended reasoning processes.

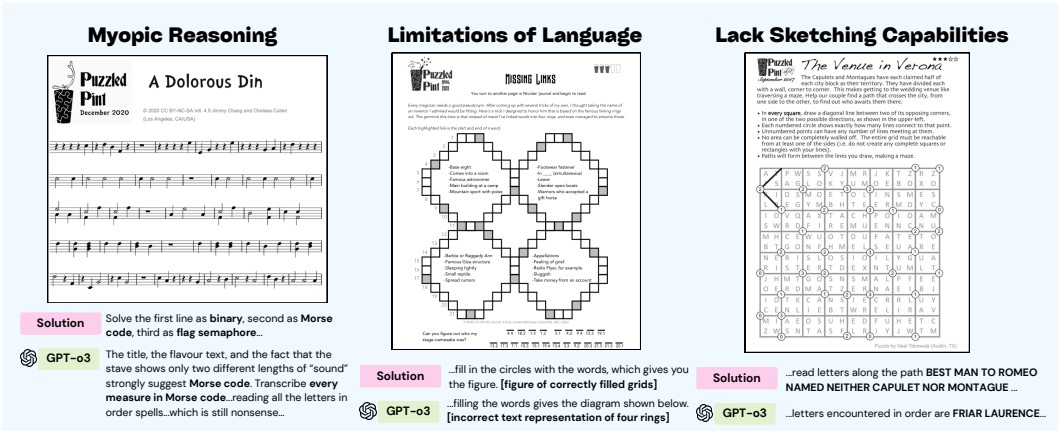

Figure 6: **Example puzzle errors. Left:** (myopic reasoning) The model is unable to backtrack when it hits a dead end. **Middle:** (language bottleneck) The model misrepresents the visual contents due to inherent limitations of texts. **Right:** (sketching errors) The model fails to execute the visual sketching steps to obtain correct intermediate outputs. High-resolution images are in Appendix E.3.

**Myopic reasoning.** Despite strong performance on conventional benchmarks, frontier models often exhibit *myopic commitment* in their reasoning. Rather than exploring alternatives or revisiting prior steps, models tend to fixate on early, surface-level hypotheses, resulting in reasoning trajectories that are locally coherent but globally misaligned with the puzzle. For example, in Figure 6, solving the puzzle requires interpreting musical notes using a mix of binary, Morse code, and flag semaphores. Instead, GPT-o3 identifies a Morse code reference early on and rigidly adheres to it–even as contradictions arise–demonstrating a lack of backtracking and verification.

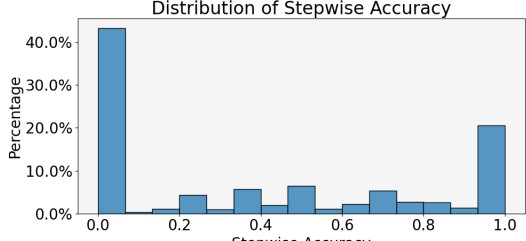

Figure 7: **Stepwise accuracy distribution of GPT-o3.** GPT-o3 receives stepwise accuracy of 0 for most puzzles, highlighting the model's myopic reasoning tendencies and its inability to backtrack after committing to an incorrect first step.

To further examine this behavior, we analyze the stepwise accuracy distribution of GPT-o3 (Figure 7). We find that, on most puzzles, the model receive a score of 0, meaning the model often fails to correctly identify even the first step of the solution. Once committed to an incorrect path, the model rarely recovers, highlighting its brittle reasoning and a lack of dynamic self-correction, especially when it cannot rely on external environments for verification.

**Limitations of language.** Modern multimodal models rely heavily on language-based reasoning strategies, such as chain-of-thought and code generation. However, this dependence becomes a bottleneck in puzzles with complex visual structure. In Figure 6, the puzzle is composed of four interlocking loops arranged in a clover-like pattern. This layout is visually intuitive, but difficult to represent in text.

While GPT-o3 correctly solves the word clues, it fails to capture the layout when converting the puzzle into text, as shown in Figure 8. This ultimately leads the model to derive an incorrect answer. This example highlights a broader limitation: when faced with highly complex struc-

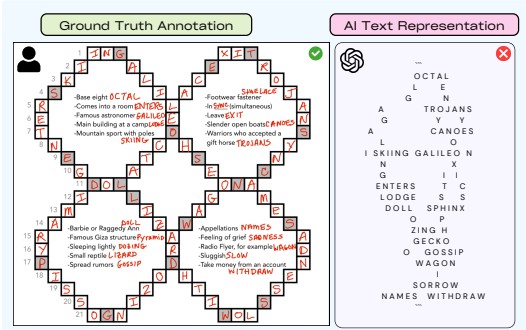

Figure 8: **Limitations of text**. An example failure case where GPT-o3 fails to represent a complex structured puzzle into text.

tured inputs, models that default to textual reasoning often lose critical spatial information. This inherent mismatch between visual intuition and language-centric inference poses a fundamental challenge to models, especially those that depend on textual or code-based reasoning chains.

**Multimodal reasoning needs sketching.** While frontier models have made notable progress in logical deduction and arithmetic reasoning, they consistently underperform on spatial tasks that require sketching, drawing, and manipulating visual structure, such as decoding based on spatial arrangements or tracing paths through grids and mazes. In Figure 6, the model correctly solves the individual clues in a grid-based puzzle but fails to trace the intended path, resulting in an incorrect final answer. Humans naturally rely on sketching or mental imagery to reason through such spatial challenges, using external or internal visualizations to keep track of evolving structure. The absence of such capabilities in current models reveals a critical gap: without the ability to sketch and update a persistent visual representation, models are prone to failure in tasks that depend on spatial coherence.

Figure 9: **Reasoning skills of failed steps.** We annotated the bottleneck steps with their reasoning skills, and we identify spatial reasoning as a primary bottleneck.

To understand the impact of sketching to model performance, we manually analyzed 30 puzzles where GPT-o3 produced incorrect answers. For each failure, we annotated the reasoning step responsible for the error with its corresponding reasoning skill. As shown in Figure 9, we found that 53.33% of these bottleneck steps involved spatial reasoning or sketching-related capabilities. This highlights a gap in models' ability to manipulate visual structure during inference. Incorporating sketch-like visual memory and reasoning (Wu et al., 2024; Hu et al., 2024; Chen et al., 2025b) may offer a promising direction toward more robust and spatially grounded reasoning AI.

## 6    CONCLUSION

This paper introduced PUZZLEWORLD, a benchmark of 667 real-world puzzlehunt-style problems for evaluating open-ended multimodal reasoning under ambiguity. PUZZLEWORLD preserves puzzles in their original formats and provides rich annotations–including modality and skill labels, as well as human-written step-by-step solution traces with intermediate states–enabling both final-answer benchmarking and fine-grained diagnosis of where and how models fail on complex, generalist reasoning tasks.

Our experiments show that, despite strong performance on conventional reasoning benchmarks, frontier models still struggle in this discovery-driven setting: accuracy drops sharply with difficulty and modality complexity, and stepwise evaluation reveals frequent early mistakes and limited backtracking. We further show that training on annotated reasoning traces can substantially improve reasoning behavior, and that these gains transfer to other visually grounded reasoning tasks. In conclusion, PUZZLEWORLD offers both a challenging benchmark and a valuable data resource. We hope that PUZZLEWORLD can foster building more robust, general-purpose multimodal reasoning models that can excel in open-ended scenarios in the real-world, such as scientific discovery, exploratory data analysis, or investigative problem-solving.

## 7    ETHICS AND REPRODUCIBILITY STATEMENT

This research focuses on developing a benchmark to support the creation of models with robust open-ended, multistep, multimodal reasoning. All data sources are cited and employed within the scope of their intended use and applicable copyright licenses. To promote transparency and reproducibility, we provide detailed data collection and annotation process in Section 4, evaluation setup in Section 5.1, and compute details in Section F of the appendix. We publicly released the PUZZLEWORLD benchmark and code to facilitate reproducibility and further research.

## 8    ACKNOWLEDGEMENT

MT is supported by the National Science Foundation (NSF) under Grant No. 2141064. Any opinion, findings, and conclusions or recommendations expressed in this material are those of the authors and do not reflect the views of the National Science Foundation. We thank the MIT Office of Research Computing and Data (ORCD) for support through ORCD Seed Fund Grants. We also thank the NVIDIA Academic Grant Program for GPU support.

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

## A  LIMITATIONS AND BROADER IMPACT

To ensure consistency and standardization across the dataset, we excluded puzzles involving under-explored or difficult-to-represent modalities such as audio, video, or interactive file-based inputs. As a result, PUZZLEWORLD may not fully capture the breadth of sensory and interaction-based reasoning found in some real-world, more challenging puzzlehunts. We discuss the potential incorporation of more challenging puzzles in Appendix C. Additionally, unlike Wang et al. (2025) that uses human annotators to transcribe textual and visual components separately, we preserve the puzzle content in its original image format and focus annotation efforts on intermediate reasoning traces. While this allows PUZZLEWORLD to provide richer annotation of the solution reasoning process, it may also introduce variability in model performance depending on the quality of their OCR capabilities. We discuss whether OCR is a limiting capability for frontier models in Appendix D.2.

Finally, our evaluation pipeline relies on LLM-based judges to automatically assess generated reasoning traces. To address this, we adopted careful prompting and cross-checking. For example, we enforce that each individual annotated step is evaluated separately by the LLM judge to determine whether it matches the model generated solution. We tested the alternative approach, where the LLM judge is prompted with the full candidate solution and outputs the latest ground truth step that the candidate response achieved. However this approach was more prone to hallucinations, as LLM judge sometimes outputs a stepwise accuracy greater than 1. As such, our approach of running the LLM judge to output a boolean on each ground truth step helps mitigate potential hallucination. Nevertheless, we acknowledge that the use of LLM-based evaluations may be subject to instability or bias, and the metrics should be taken with caution.

One possible concern with our grading scheme is that a model might "hallucinate" the correct final answer without engaging in proper reasoning. However, such a case is extremely rare in PUZZLEWORLD. The high-quality, human-designed puzzles are deliberately constructed to discourage superficial guessing, and even experienced human solvers cannot easily infer the answer without following the intended logic. In a thorough manual inspection of models' puzzle responses, we did not find any case where the model arrived at the final answer without demonstrating the necessary reasoning. In Appendix D.3, we discuss whether our stepwise scoring mechanisms successfully credit alternative solution paths.

Our goal in releasing PUZZLEWORLD is to advance research in general-purpose, multimodal reasoning systems. However, we recognize that increasingly capable AI models, especially those skilled at complex reasoning, carry risks of misuse. These include the potential for externalizing or replacing human reasoning in settings where authenticity or creativity is essential, such as education, scientific authorship, or collaborative problem-solving. While our dataset does not pose direct risks on its own, we support future work that includes safeguards to mitigate misuse and encourages the responsible deployment of reasoning-capable AI systems in alignment with human values.

## B  USE OF LARGE LANGUAGE MODELS

This project used Large Language Models (LLMs) to assist dataset verification and model evaluations. Details are described in the Sections 4.3 and 5.1. We also acknowledge the use of LLMs to assist with correcting grammatical errors and improving clarity of the writing. This assistance was limited to language refinement and did not affect the core methodology, scientific rigour, or originality of the research. We confirm that no AI-generated content has been presented as our own intellectual contribution.

## C  FUTURE DIRECTIONS

We note that some prior benchmarks (Chen et al., 2025a) adopt a *structured reflective reasoning* paradigm, centered on structured, text-based puzzles with explicit rules, enabling precise state verification and surgical analysis of multi-step logical reasoning. In contrast, PUZZLEWORLDemphasizes multimodal, discovery-driven reasoning in which the puzzle rules are implicit and must be inferred by the solver. This setting offers a more holistic evaluation of generalist reasoning capabilities that requires a broader range of cognitive skills, though at the cost of less precise, LLM-approximated intermediate state verification due to the absence of explicit rule structures.

An exciting direction for future work is to extend PUZZLEWORLD with structured, verifiable reflective-reasoning frameworks akin to those in Chen et al. (2025a). However, we caution that PUZZLEWORLD's open-ended and multimodal puzzles often involve diverse, creative reasoning steps that are difficult to formalize as explicit rules, making it challenging to achieve the same level of rule-based granularity as in more structured puzzle domains in Chen et al. (2025a). That said, integrating reflective reasoning paradigms with PUZZLEWORLD could yield richer diagnostic tools and may ultimately benefit the training and evaluation of frontier reasoning models.

We acknowledge that, beyond PuzzledPint, extensive puzzlehunt-style puzzle sources could be integrated to expand PUZZLEWORLD, such as MIT Mystery Hunt (MIT, 1981–). While MIT Mystery Hunt is an attractive source of challenging, high-quality puzzles, we note several considerations that make it challenging to integrate it to PUZZLEWORLD's initial release.

- Unlike PuzzledPint (released under Creative Commons), MIT Mystery Hunt puzzles do not have a centralized copyright policy. Permissions would need to be obtained from individual authors, which adds substantial overhead.

- Mystery Hunt puzzles frequently involve audio, video, interactive web tools, custom code, and even physical components. These are exciting modalities for future generalist AI systems, but they are difficult to standardize under PUZZLEWORLD's stepwise annotation format and are not well supported by current frontier models.

- Mystery Hunt puzzles are intentionally designed to challenge top teams and often require multiple people to solve each puzzle. Given that frontier models already struggle on the entry-level PuzzledPint puzzles, including much harder Mystery Hunt puzzles would likely provide limited practical diagnostic value at this stage.

Therefore, for the present, we prioritize the more accessible PuzzledPint puzzles, as they are both suitably challenging and sufficiently structured to support reliable stepwise annotation and informative intermediate diagnostics. That said, as models and annotation tools improve, future work that extends PUZZLEWORLD to incorporate more advanced modalities and higher-difficulty sources – potentially including Mystery Hunt puzzles – would offer a richer evaluation suite.

## D    ADDITIONAL DISCUSSION

### D.1    ON THE ABSTRACT NATURE OF PUZZLEHUNTS

We note that PUZZLEWORLD, like prior puzzle datasets (Chia et al., 2024; Ghosal et al., 2024; Estermann et al., 2024), is abstract in nature rather than drawn from real-world tasks such as mathematics or physics. While this means that PUZZLEWORLD does not directly evaluate a model's ability on any specific application, this abstraction is intentionally designed to assess generalist, open-ended reasoning while reducing pattern memorization. For example, prior work has shown that symbolic variants of math datasets can reveal brittle reasoning behavior in models that otherwise appear strong (Mirzadeh et al., 2025). PUZZLEWORLD's abstraction thus helps isolate and measure a model's generalist and genuine reasoning ability beyond domain-specific patterns.

Existing reasoning benchmarks – ranging from math and coding tasks – typically operate in closed-ended, well-defined environments. These datasets assess correctness under fixed rules, but they do not require models to navigate underspecified or open-ended problem spaces. In contrast, many real-world tasks (e.g., exploratory data analysis, investigative research, science discovery) involve ambiguous signals, multiple possible solutions, and iterative hypothesis testing.

PUZZLEWORLD fills this gap in two ways that previous abstract reasoning benchmarks do not. First, unlike traditional puzzle datasets (e.g. Sudoku) that evaluate within a limited rule set, our puzzlehunt puzzles require leveraging diverse reasoning competencies to dynamically form and evaluate hypotheses from nuanced multimodal signals. Second, our manually annotated reasoning traces enable systematic analysis of model behavior in these open-ended settings, revealing characteristic failure modes of frontier models such as "myopic commitment" (Section 5.4). Our transfer experiments in Table 4 has also shown that capabilities exercised in PUZZLEWORLD transfer to real-world benchmarks, suggesting that puzzlehunt-style reasoning is a useful proxy for generalist reasoning. Nonetheless, we acknowledge that no single benchmark can represent all forms of reasoning.

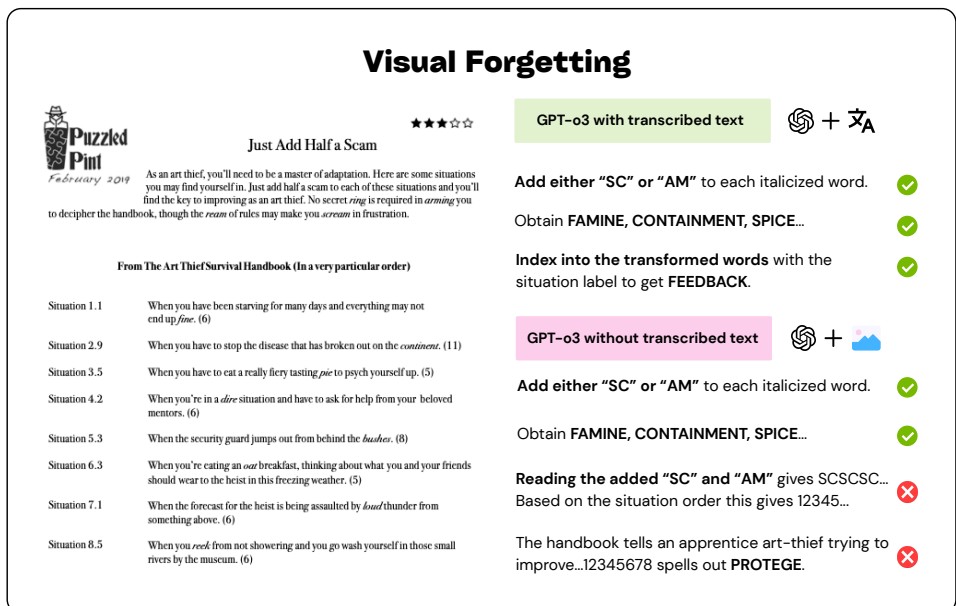

Figure 10: **Example error of visual forgetting.** GPT-o3 failed to correctly reference the text clues in the puzzle image, but it solved the puzzle once the clues were provided in transcribed text form.

## D.2 ARE REASONING MODELS BOTTLENECKED BY TEXT RECOGNITION?

We acknowledge that multimodal large language models may be bottlenecked by text recognition in images. In the construction of PUZZLEWORLD, since Wang et al. (2025) has shown that today's best multimodal reasoning models are not primarily constrained by optical character recognition (OCR) capabilities, we devoted our manual effort to construct fine-grained annotations. Although each puzzle includes a human-transcribed "flavor text", this may not capture all textual content.

We thus provide an analysis on whether OCR is limiting the performance of frontier models on PUZZLEWORLD. We selected a 5% subset of strictly text-only puzzles, manually transcribed all textual content, and compared GPT-o3's performance with and without the full transcription. A paired t-test showed no significant difference ($t = 0.52$, $p > 0.1$), suggesting that text recognition limitations are not a major bottleneck for the frontier multimodal models.

That said, we did observe isolated cases where GPT-o3 improved from partial to full solutions once transcription was provided. Upon examination, these instances appear to stem from model hallucination in later stages of reasoning when relying only on images, whereas the transcription helps anchor its reasoning more reliably. For example, in Figure 10, GPT-o3 solves the puzzle correctly when given the transcribed text. When relying solely on the image, GPT-o3 can solve the intermediate text clues but fails during extraction: it misreferences the visual text, as shown by its inability to recover the added letters against the italicized words in the image, and hallucinates a final answer only tangentially related to the flavor text. This aligns with patterns of visual forgetting observed in prior works (Sun et al., 2025).

## D.3 CAN PUZZLEWORLD'S STEPWISE SCORING CREDIT CREATIVE SOLUTIONS?

Due to the PUZZLEWORLD's open-ended nature, a legitimate concern is whether PUZZLEWORLD's stepwise scoring mechanism can fairly credit creative, alternative reasoning paths, rather than being limited to the annotated reference chain. In puzzlehunts, the solution steps are intentionally designed to be interlocking and sequentially dependent. It is thus hard to reach the correct answer while following a completely different solution path, and correct partial chains must partially converge with the annotated ground-truth, even if the solver temporarily deviates or makes logic leaps.

That said, human puzzlehunters do commonly make educated guesses that skip over intermediate steps. For example, after identifying three of the four final answer letters ("I _ L R"), a solver might correctly infer "ICLR" without fully completing the preceding clue. PUZZLEWORLD's step-level scoring explicitly accounts for this behavior: we identify the latest ground-truth step that appears in the candidate solution. In this scenario, because the solver correctly arrives at the final answer, our scoring grants full credit, even if several intermediate steps were omitted or approximated.

Table 5: **GPT-o3's performance on PUZZLEWORLD by difficulty.** GPT-o3's accuracy monotonically decreases as difficulty increases.

| Difficulty | Accuracy | Stepwise |
|---|---|---|
| Easy | **18.46%** | **40.58%** |
| Medium | 14.76% | 39.38% |
| Hard | 9.64% | 39.71% |

Table 6: **GPT-o3's accuracy on PUZZLEWORLD by difficulty and modality.** Within each modality, GPT-o3's accuracy consistency decreases with increased difficulty, with a relatively well performance on easy structured puzzles.

| Difficulty | Text | Visual | Structured |
|---|---|---|---|
| Easy | **19.5%** | **11.9%** | **20.4%** |
| Medium | 15.3% | 8.9% | 13.5% |
| Hard | 10.6% | 7.6% | 9.6% |

As a proxy for evaluating how robust our step-level crediting is to alternative partial chains, we randomly sampled 10% of PUZZLEWORLD and automatically paraphrased GPT-o3's solutions using GPT-5-mini. We then compared the LLM-judge's step-level scores on the original and paraphrased solutions. We observe strong positive correlation between the two sets of scores ($r = 0.885$, $p < 0.001$) with a mean absolute error of 0.066, indicating a high degree of score stability across semantically equivalent but structurally varied solutions. Although this does not exhaustively cover all possible reasoning paths, it indicates that the LLM-judge can reliably credit alternative solutions.

# E PUZZLEWORLD DETAILS

## E.1 CHECKING FOR DATASET CONTAMINATION

To assess the possibility of data contamination, we test whether GPT-o3 (OpenAI, 2025) has memorized any of the puzzles in our dataset. Specifically, inspired by prior work Tanzer et al. (2023); Chi et al. (2024), we prompt the model to reconstruct the flavor text for 40 randomly sampled puzzles out of the 84 that were answered correctly. We then use GPT-4o (Achiam et al., 2023) to automatically evaluate the similarity between the reconstructed and original flavor texts. We find a reconstruction accuracy of 0%, suggesting little to no evidence of data leakage. Furthermore, since Puzzled Pint (Puzzled Pint, 2025) publishes new puzzles on a monthly basis, our dataset can be continuously updated to mitigate the risk of model overfitting on released content.

## E.2 DIFFICULTY LABEL ANALYSIS

We acknowledge that the difficulty labels in PUZZLEWORLD are obtained from the original PuzzledPint sources, without manual calibration. According to the PuzzledPint website (Puzzled Pint, 2025), all submitted puzzles undergo internal playtesting by an editorial team, where the difficulty tags are revised and finalized by expert editors and reflect a community-standard difficulty assessment. We thus treat them as expert annotations rather than ad-hoc metadata.

To understand whether the human-labelled difficulty labels meaningfully reflect task difficulty, we inspect GPT-o3's performance broken down by difficulty in Table 5. As expected, we observe that GPT-o3's accuracy decreases as difficulty level increases. Another valid concern is that human-perceived difficulty and AI-perceived difficulty might not align. For example, certain diagram-heavy puzzles may be straightforward for humans but require nontrivial spatial reasoning for AIs. To investigate this, we further break down GPT-o3's performance by modality and difficulty.

As observed in Table 6, within each modality, accuracy consistently decreases with difficulty. Notably, GPT-o3 performs relatively well on easy structured puzzles – likely due to their clean, regular layouts – but its performance drops sharply on harder structured puzzles where the diagrams become more irregular and visually complex. Overall, both aggregated and modality-level results show that PuzzledPint's difficulty labels provide a meaningful and consistent difficulty metric for AI models.

### E.3 PUZZLEWORLD IMAGE SAMPLES

We provide high-resolution images of puzzle samples used in this paper.

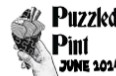 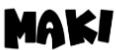 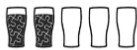

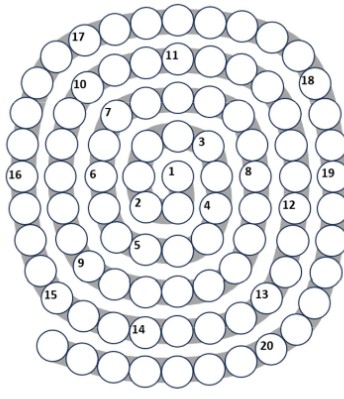

*Location recap: Stephanie mentioned to her roommate Katie that she didn't know there were multiple types of sushi. Upon hearing this, Katie decided to teach Stephanie about the many different forms that sushi can take!*

At a sushi restaurant, Katie continued Stephanie's crash course in sushi.

"We'll start with the basics," Katie said. "You mentioned you have only had California rolls before. Those fall into the category of *maki*, or sushi rolls. There's seaweed and rice on the outside of the maki sushi, and the filling is in the center. To make maki, chefs lay out all the components on top of each other so that they **overlap to** an extent, and then tightly roll them into a circle."

"Wow, I see!" Stephanie said. "This particular piece of maki is enormous... Here, let's split it **down the middle** and share it. Say, do you know why this maki is so large?"

"I'm not sure," Katie replied. "But we'll _____________!"

Clues:

1. Former bride
2. Sworn allegiance
3. Descriptor for blood
4. Pluckable flower part
5. A Greek letter
6. A feeling more intense than dislike
7. Psychic communicator, e.g., Professor X
8. Considerate
9. "Pen-" follower
10. One of three from a famous trio
11. Group of musicians led by a conductor
12. Poe bird
13. E.g., Sun-grid, combustion, or Unity
14. E.g., optic or tibial
15. In geometry, a point where two or more lines meet
16. For ____ (as an illustration)
17. Folkloric Irish creature
18. Rally around a common cause
19. Maryland's state reptile
20. Shorter alternative to a signature

Stephanie Yang (Boston, MA)

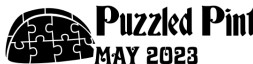 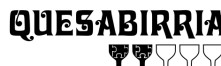

More akin to quesadillas than tacos, quesabirria tacos feature stewed, shredded beef, goat, or jackfruit melted together with cheese inside of a tortilla. The tortilla is folded with two **nearly identical halves** pressed together. These tacos are often served with *a b*roth or *c*onsomme for *d*ipping *e*ach taco. In fact, you'd be hard-pressed to find a more important part of the quesabirria experience than the long-simmered, **well-reduced** broth.

Assertive and reckless (5)
Assistance (3)
Attempt to overfill (4)
Bountiful celebratory meal (5)
Classification of vinegar or citrus juice (4)
Currency in London (5)
Euphemism for ocean (4)
Ground covering that is greener where you aren't (5)
High fat dairy liquid (5)
Kind and polite (4)
Launderer's woe (5)
Man who wrote about a raven (3)
Noted Vatican resident (4)
Okie doke (3)
Quick (4)
Russian dictator (6)
Second word of a celebration in New Orleans (4)
Sibling's daughter (5)
Site for restaurant reviews (4)
Support garments (4)

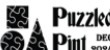 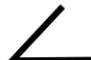 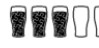

Archimedes and Benjamin Banneker are sitting on a bench, having a discussion.

Archie: Tell me old friend, when you're drawing angles, don't you just feel amazing – like your mind is the willing subject of a kidnapping to another dimension?
Benjie: Wow, Archie, I never thought of it that way. I just know they always seem to make fizzy feelings inside of me. You could say that I am one hopeless lover when it comes to angles!
Archie: You might not realize it at first glance, but if you look really closely, you'll realize it always takes three points to define an angle, and I've always liked threes: Three musketeers, three-legged races, **three-syllable words**.
Benjie: I just feel angles make so many awesome **connections**: they are what all the other shapes we draw depend on.
Archie: I know, from acute to obtuse, angles have always seemed **larger** than life!

Carl Friedrich Gauss walks up and tries to join the discussion.

Carl: Hey I like drawing angles too.
Archie: Um, excuse you! This was an A B conversation, so you can C your way out!

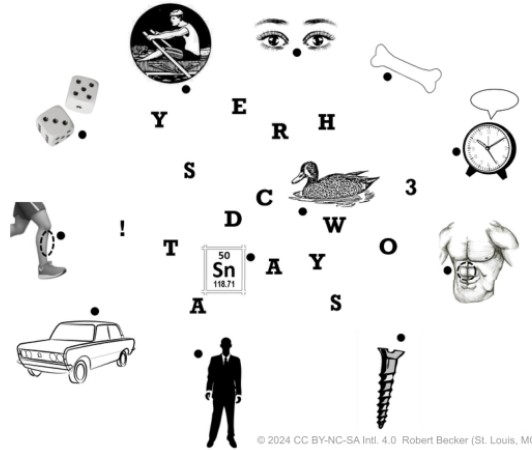

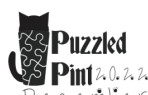 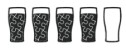

Mittens has been forcefully relocated by his hoomans to a new mansion. He now faces the daunting task of exploring his new home. To further complicate matters, the mansion has four levels, and his perspective will have to keep **shifting** with each new level. Along the way, Mittens will discover the **shortest** path that leads him to his true heart's desire, for which he will have to make his way all the way up onto the roof.

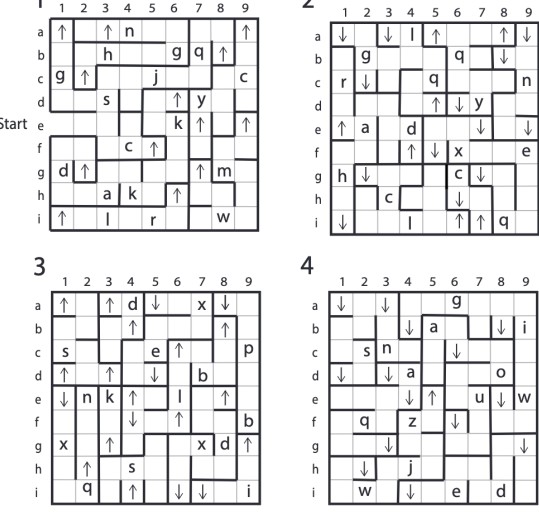

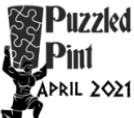

# EROS'S ARROWS

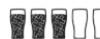

APRIL 2021

Eros, the god of love, has shown up at a speed-dating session. Each person at this speed-dating session is assigned a **number**. Eros used his arrows to make each person fall in love with someone on the other side. Eros is mischievous: he caused everyone to fall in love with one person, everyone to be loved by one person, but no one's love to be reciprocated.

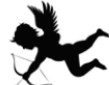

Eros has many arrows, but only five different arrowheads; each arrowhead does something different. Additionally, Eros may shoot up to three arrows **from** each person to get the job done, even repeating a type if required.

Can you figure out who fell in love with whom?

**Possible arrowhead effects:**
• Add the number of letters in the person's name
• Divide by 3
• Multiply by 2
• Subtract 2
• Subtract 5

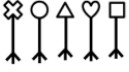

1. Blake  △⊠  •
2. Charlie ⊠○△•
3. Avery  △△♡  •
4. Kerry  △○△  •J
5. Parker  ⊠♡  •

X        A        C
   B
E        D    F
   G    H
      Z
I   L   M      T
   K   W   O   N
U          Y   S
   Q    R
P   V        I

• 6. Leslie □⊠○
• 7. Jesse○
• 8. Emerson♡♡
• 9. Jamie □♡
• 10. Alex ○

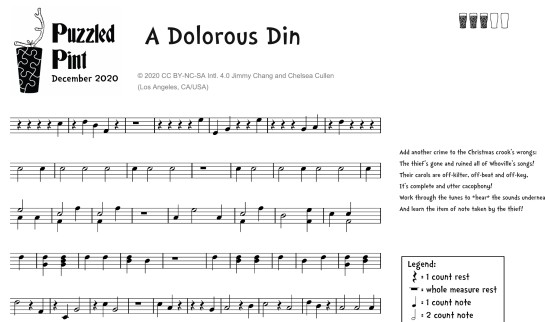

## A Dolorous Din

Puzzled Pint
December 2020

Add another crime to the Christmas crook's wrongs:
The thief's gone and ruined all of Whoville's songs!
Their carols are off-kilter, off-beat and off-key,
It's complete and utter cacophony!
Work through the tunes to "hear" the sounds underneath,
And learn the item of note taken by the thief?

Legend:
♪ = 1 count rest
— = whole measure rest
♩ = 1 count note
♩ = 2 count note

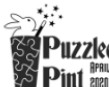

## MISSING LINKS

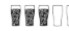

You turn to another page in Nicolas' journal and begin to read.

*Every magician needs a good pseudonym. After coming up with several tricks of my own, I thought taking the name of an inventor I admired would be fitting. Here's a trick I designed to honor him that is based on the famous linking rings act. The gimmick this time is that instead of metal I've linked words into four rings, and even managed to entwine those.*

*Each highlighted link is the start and end of a word.*

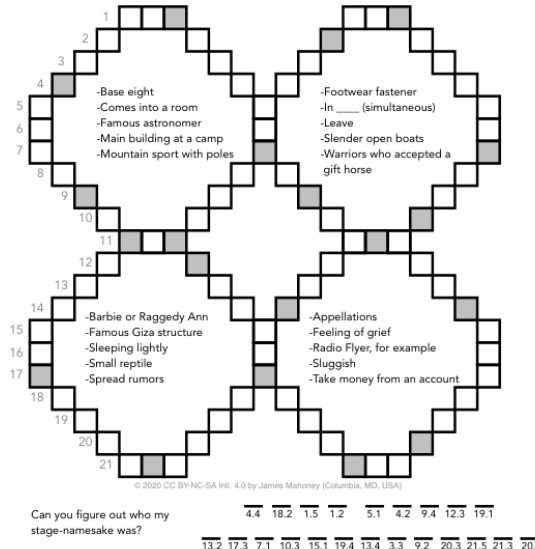

- Base eight
- Comes into a room
- Famous astronomer
- Main building at a camp
- Mountain sport with poles

- Footwear fastener
- In _____ (simultaneous)
- Leave
- Slender open boats
- Warriors who accepted a gift horse

- Barbie or Raggedy Ann
- Famous Giza structure
- Sleeping lightly
- Small reptile
- Spread rumors

- Appellations
- Feeling of grief
- Radio Flyer, for example
- Sluggish
- Take money from an account

Can you figure out who my stage-namesake was?

4.4  18.2  1.5  1.2    5.1  4.2  9.4  12.3  19.1

13.2  17.3  7.1  10.3  15.1  19.4  13.4  3.3  9.2  20.3  21.5  21.3  20.1

---

## The Venue in Verona ★★★☆☆

*September 2017*

The Capulets and Montagues have each claimed half of each city block as their territory. They have divided each with a wall, corner to corner. This makes getting to the wedding venue like traversing a maze. Help our couple find a path that crosses the city, from one side to the other, to find out who awaits them there.

- In **every square**, draw a diagonal line between two of its opposing corners, in one of the two possible directions, as shown in the upper-left.
- Each numbered circle shows exactly how many lines connect to that point.
- Unnumbered points can have any number of lines meeting at them.
- No area can be completely walled off. The entire grid must be reachable from at least one of the sides (i.e. do not create any complete squares or rectangles with your lines).
- Paths will form between the lines you draw, making a maze.

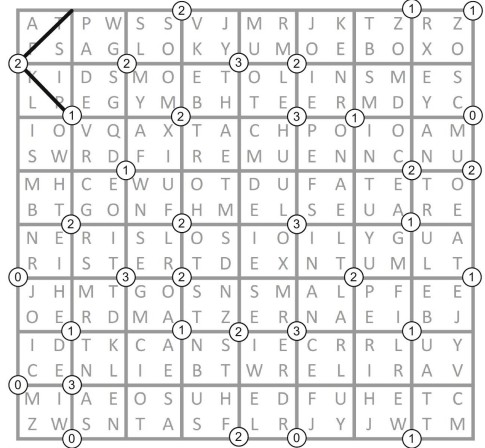

Puzzle by Neal Tibrewala (Austin, TX)

# F    BENCHMARKING DETAILS

## F.1    COMPUTE RESOURCES

All evaluations and experiments in this paper were conducted on a remote cluster equipped with two NVIDIA H200 GPUs (each with 141 GB HBM3 memory). Runtime for each model evaluation varied between 10–48 hours depending on the model size and architecture.

## F.2    SUPERVISED FINETUNING

All fine-tuning experiments were conducted using the LLaMA Factory framework (Zheng et al., 2024). For fine-tuning on the 8B InternVL3 model, we used LoRA fine-tuning with a rank of 8, a learning rate of $1 \times 10^{-6}$, and trained for 3 epochs on PUZZLEWORLD. For the transfer experiments, we fine tune a Qwen2.5-VL-7B-Instruct model, using full parameter finetuning with a learning rate of $1 \times 10^{-5}$ for 5 epochs on PUZZLEWORLD. We did not perform extensive hyperparameter tuning for any of these experiments.

## F.3    HUMAN BASELINE DETAILS

We provide additional context for the human baseline in Table 2. While the human baseline only covers a subset of PUZZLEWORLD, the resulting baselines reveal clear separations between human tiers and current models—for example, a >30% gap between human enthusiasts and GPT-o3. This gap is substantially larger than the uncertainty introduced by the sample size, suggesting that the current human results provide a reliable directional reference point.

Given our resource constraints, we prioritized depth and quality in the human evaluation rather than exhaustiveness. To reduce individual variability in a task as open-ended as PuzzleWorld, each sampled puzzle was solved by at least two independent participants, each allotted up to one hour and asked to produce detailed reasoning traces. Since we assign multiple participants to each puzzle to ensure consistency, each puzzle modality has more than 40 human attempts, which we consider sufficient for a modality-level human baseline.

Finally, we note that the same LLM-judge protocol is applied to both human and model solutions, ensuring that the comparison is fair and step-symmetric. We observe that, unlike AI models, visual and structured modalities do not pose additional difficulty for human solvers, further highlighting the multimodal limitations of current foundation models.

## F.4    ROBUSTNESS OF LLM-JUDGE TO IMPERFECT ANNOTATIONS

PUZZLEWORLD contains extensive manual annotation to provide fine-grained analysis on models' performance. In this section, we include an ablation on the impact of potential human annotation noise on our step-level grading accuracy. To simulate annotation noise, we select 10% of PUZZLEWORLD and use GPT-5-mini to paraphrase each step of the ground truth annotation. Then, we evaluate GPT-o3's output on both the original and paraphrased ground truth. We observe strong positive correlation between the two sets of scores ($r = 0.898$, $p < 0.001$) with a mean absolute error of 0.056. This indicates that PUZZLEWORLD's stepwise evaluation pipeline is robust to moderate noise in the stepwise annotations.

## F.5    PROMPT FOR BENCHMARKING

Below is the system prompt template for benchmarking models on PUZZLEWORLD, which is adapted from Wang et al. (2025).

```
You will be presented with a puzzle to solve. The puzzle may not have specific instructions,
but you know that the answer to the puzzle is a word or short phrase (or rarely, a number).

Do not ask any questions about how to proceed, just do your best to solve the puzzle.
Here are some tips for solving puzzles of this type:

General Tips:
- Puzzles will often have multiple steps to get to the answer word. You can usually tell you
are on the right track if the intermediate answers agree with the title, flavor, or theme
of the puzzle.
- You can usually find hints in the introductory text. For example references to "in the dark"
or "sight" are often hints something is encoded with braille.
- Puzzles often incorporate acrostics: a clue where the first letter, syllable, or word of
each line, paragraph, or other recurring feature spells out a word or message.
- If you end up with a garbled "alphabet soup", then look for a clue on how to order them.
- Indexing is one of the most common puzzle mechanisms. Try indexing when you have a list of
words or phrases and a corresponding list of numbers. Count into the word or phrase by the
given number and record the letter in that position. For example: "2 Cake, 6 Pudding, 5
Shortening" gives you "ant".
```

```
- Alpha-numeric codes are also very common. If you end up with a list of numbers try replacing
the numbers with the corresponding letters like this: 1 = A, 2 = B, 3 = C... 26 = Z.
Occasionally, these types of codes will "wrap around", so don't despair if you see a
number greater than 26. Just subtract 26 and try again. In this scenario 27 (27-26 = 1) =
A, 28 (28-26 = 2) = B etc. If you try this and it doesn't work, try other numeric codes
such as ASCII.
- Often a puzzle repeats a strategy multiple times.

You will likely need to backtrack frequently, so make sure to write out your steps as you go.
If you get stuck, try to think of a new way to approach the puzzle. Try:
- Rereading the title and the flavor text. These are the most important hints about what type
of strategies, themes or cultural references might be used to solve the puzzle.
- Checking for pop culture references
- Checking for references to a song/poem/book/movie/TV show

For strings, examples of strategies you might try include:
- Alphabetizing
- Using leftover letters to spell something
- Rearranging the letters (aka anagrams or "transposing")
- Seeing if there are any acronyms
- Diagonalizing (taking the first letter of the first answer, the second letter of the second
answer, etc.)
- Looking for unusual letter frequencies
- Puns and homophones
- Shifting from letters to numbers

For numbers, try:
- Shifting from numbers to letters
- Using it as a phone number
- Treating numbers as dates
- Treating numbers as ASCII numbers
- Seeing if there are any strange sequences
- Seeing if prime numbers are involved

For images, try:
- Looking at it in a mirror
- Squinting at it from far away
- Tilting it
- Looking at it upside down
- Looking through it
- Transcribing it neatly
```

We additionally append the user prompt:

```
Your task is to solve the following puzzle. The attached images are presented in the order
they are referenced in the text.

The puzzle's title is: {}
The puzzle's flavor text is: {}

---
Write out a step-by-step solution to the puzzle. At the end of your solution, write your
answer in the following format:
Answer: <answer>
```

Below is the prompt for LLM judge:

```
Answer Equivalence Instructions:
Using the puzzle and the reference solution, grade the candidate solution as follows.

For every reasoning step of the reference solution, output True if the candidate solution both
includes
the step and achieves the same intermediate result of the step, otherwise False.
Explain why the candidate's solution did or did not get the reasoning step correct.
Do not add more steps than there are in the reference solution and evaluate every step
in the reference solution.
There is a exception in scoring for the last reasoning step. Identify the candidate output
solution.
If the candiate output solution is the exact same as the reference solution answer of
\"{puzzle_solution}\",
then output final step as true.
```

# G ANNOTATOR DETAILS

We employ university undergraduates to assist the human annotation process in PUZZLEWORLD. All annotators are compensated at a rate of $16.00 per hour. Prior to annotation, annotators receive detailed guidelines and participate in training sessions to ensure annotation consistency and task understanding.

## G.1 ANNOTATOR INSTRUCTIONS

We provide the instructions given to annotators below:

```
# Instructions for Submitting a Puzzle
```

```
To submit a puzzle, fork this repository and create a new branch. Then, create a new folder
`{puzzle_name}` in the `data/puzzles` folder, and place the following files in it:

- `metadata.json`: A JSON file containing the metadata of the puzzle
- `content.png`: The image of the puzzle content
- `figure_{N}.png`: (Optional) Figures illustrating the reasoning steps

For an example puzzle, see the `data/puzzles/example` folder. After you are done, create a pull
request to merge your branch into the main repository.

Note, please replace any spaces in the puzzle name with `_` when creating the new folder!

## Metadata
The `metadata.json` file should contain a JSON object with the following fields:

| Field Name  | Type   | Description                                       |
|-------------|--------|---------------------------------------------------|
| title       | string | The title of the puzzle                           |
| flavor text | string | The flavor text of the puzzle, possibly empty     |
| difficulty  | string | The difficulty level of the puzzle (easy, medium, hard) |
| solution    | string | The solution to the puzzle                        |
| reasoning   | Step\[ \] | An ordered list of reasoning [steps](#reasoning-step) towards the
solution |
| modality    | string\[ \] | A list of input [modalities](#a-list-of-input-modalities) the puzzle
contains |
| skills      | string\[ \] | A list of [skills](#a-list-of-reasoning-skills) required to solve the
puzzle |
| source      | url    | Thel link to the puzzle                           |

### Reasoning Step
The `reasoning` field should contain a list of `Step` objects, which are represented as
dictionaries with the following fields:
| Field Name  | Type      | Description                                       |
|-------------|-----------|---------------------------------------------------|
| explanation | string    | The textual explanation of the step              |
| figure      | file path | (Optional) File path to a figure illustrating the step |

Each of the explanation should begin with one of the following atomic actions:
- Pattern discovery: discover patterns / insights from current information
  - E.g. discovering that current laser patterns are semaphores
- Sketching: sketching on or interacting with visual elements
  - E.g. traversing through a maze
  - E.g. connecting the dots
- Manipulation: manipulating or arranging a sequence of elements
  - E.g. sorting alphabets in order
  - E.g. applying cryptic encoding / decoding
- Combining / Chaining: combining or chaining multiple pieces of observations
  - E.g. matching patterns in images with text segments
- Extraction: extracting information from one pattern or observation
  - E.g. extracting letters from semaphore patterns

(Note: the exact wording of action is not important as long as it resembles one of the above
categories)

Each explanation step should consist of one action and the intermediate outcome of the action e.g.
Identify the pattern that (...), which is (...)

### A List of Input Modalities
| Keyword      | Description                                                  |
|--------------|--------------------------------------------------------------|
| `text`       | Textual information                                          |
| `visual`     | Unstructured visual information e.g. images, icons, fonts, etc. |
| `structured` | Structured visual information e.g. tables, graphs, crosswords, etc.|

### A List of Reasoning Skills
| Keyword       | Description                                                                  |
|---------------|------------------------------------------------------------------------------|
| `logic`       | Logic reasoning e.g. rule deduction or inferring conclusion given partial
information |
| `wordplay`    | Manipulating words based on linguistic properties e.g. anagrams, homophones, etc. |
| `spatial`     | Spatial or visual understanding, manipulation and navigation e.g. mazes, connecting
dots, etc. |
| `cryptic`     | Encoding and decoding information e.g. ciphers, indexing, etc.               |
| `knowledge`   | Leveraging domain-specific knowledge e.g. history, science, etc.            |
| `commonsense` | Applying common sense reasoning e.g. physical laws, social norms, etc.      |
| `tool_use`    | Searching through an external database for information unlikely in model's training
data, such as Google Maps |
```

