# OpenReview forum: "PuzzleWorld: A Benchmark for Multimodal, Open-Ended Reasoning in Puzzlehunts"
_ICLR.cc/2026/Conference — ICLR 2026 Poster_

### Official Review · Reviewer_s7Za · 2025-10-22

**Soundness:** 3
**Presentation:** 3
**Contribution:** 3
**Rating:** 6
**Confidence:** 4

**Summary:**

PUZZLEWORLD introduces a benchmark of 667 puzzlehunt-style problems designed to evaluate multimodal, open-ended reasoning in AI systems. Unlike conventional benchmarks with well-defined tasks, puzzlehunts require discovering problem structures from ambiguous, multimodal clues, mirroring real-world challenges like scientific discovery or investigative problem-solving. The dataset includes detailed annotations of solutions, step-by-step reasoning traces, and cognitive skill labels. Evaluations show state-of-the-art models achieve only 1–14% final answer accuracy, with the best model (GPT-o3) matching novice human performance but lagging behind enthusiasts and experts. The benchmark also enables diagnostic analysis, revealing model limitations in backtracking, visual reasoning, and sketching capabilities. Fine-tuning on reasoning traces improves stepwise accuracy, demonstrating the utility of PUZZLEWORLD for advancing general-purpose reasoning.

**Strengths:**

1. PUZZLEWORLD fills a critical gap by focusing on open-ended, discovery-driven problems rather than constrained tasks. Its emphasis on multimodal clues and unstructured problem-solving aligns with real-world scenarios, providing a more holistic evaluation of reasoning capabilities.
2. The dataset includes meticulously curated step-by-step reasoning traces, cognitive skill labels, and modality tags. These annotations support fine-grained diagnostics, error analysis, and model training, surpassing existing benchmarks in depth and utility.
3. The study comprehensively tests frontier models (e.g., GPT-o3, Claude Opus) and human baselines, highlighting significant performance gaps. Stepwise accuracy metrics offer nuanced insights beyond final answers, revealing intermediate reasoning failures.

**Weaknesses:**

1. I think the experimental analyses are not sufficient. For example, we see that almost all MLLMs perform poorly, but the reasons behind this are worth further investigation. Is the model's poor performance due to poor performance in the visual modality or poor performance in the text modality? In my experience, most MLLMs are unable to perform well in downstream tasks due to their insufficient ability to recognize text in images.
2. The article lacks discussion of more related work, such as [1].
3. I look forward to seeing more case studies.

[1] LatEval: An Interactive LLMs Evaluation Benchmark with Incomplete Information from Lateral Thinking Puzzles.

**Questions:**

See the above weaknesses.

---

> ### Author Response · Authors · 2025-11-23
> **Response to Reviewer s7Za**
>
> We thank the reviewer for their detailed feedback! We appreciate the reviewer's time in writing the review, and we are happy to respond to the concerns raised.
>
> **W1. Is the model's poor performance due to poor performance in the visual modality or poor performance in the text modality?**
>
> We thank the reviewer for raising this important point, and we acknowledge that many MLLMs may be bottlenecked by text recognition in images. Although each puzzle includes a human-transcribed “flavor text”, this may not capture all textual content present. To assess whether text recognition is a limiting factor, we selected a 5% subset of strictly text-only puzzles, manually transcribed all textual content, and compared GPT-o3’s performance with and without the full transcription. A paired t-test showed no significant difference ($t = 0.52, p > 0.1$), suggesting that text recognition limitations are not a major bottleneck for the frontier multimodal models.
>
> That said, we did observe isolated cases where GPT-o3 improved from partial to full solutions once transcription was provided. Upon examination, these instances appear to stem from model hallucination in later stages of reasoning when relying only on images, whereas the transcription helps anchor its reasoning more reliably. This aligns with patterns of visual forgetting observed in prior work such as [1]. We will incorporate this analysis and caveat into the revised draft.
>
> Because annotation resources are limited, we prioritized step-level reasoning over full transcription, as it enables fine-grained error analysis. For example, in Figure 9, 53.3% of GPT-o3’s failures arise from spatial reasoning errors (primarily in visual/structured modality), while 20% are due to logic errors (often in text modality). These error types are also reflected in GPT-o3's accuracies broken down by difficulty vs. modality:
>
> | Difficulty/Modality           | Text    | Visual | Structured |
> |:---------------- |:--------------------:|:--------------------:|:--------------------:|
> | Easy | 19.5% | 11.9% | 20.4% |
> | Medium | 15.3% | 8.9% | 13.5% |
> | Hard| 10.6% | 7.6% | 9.6% |
>
> Within each modality, accuracy consistently decreases with difficulty. Notably, GPT-o3 performs relatively well on easy structured puzzles –likely due to their clean, regular layouts – but its performance drops sharply on harder structured puzzles where the diagrams become more irregular and visually complex. Taken together, these results indicate that while text recognition is not a primary bottleneck, spatial reasoning limitations in the visual modality play a significant role in the model’s overall performance. We will include this discussion in the final draft of this paper.
>
> **W2. The article lacks discussion of more related work, such as LatEval.**
>
> We thank the reviewer for pointing us to this related work. LatEval provides a text-based diagnostic dataset focused on lateral thinking through story completion, assessing a model’s ability to ask creative questions and identify non-obvious connections. In contrast, PuzzleWorld requires a combination of lateral thinking, in inferring implicit rules from puzzle content, and vertical thinking, in executing multi-step reasoning once those rules are discovered. Thus, PuzzleWorld exercises a broader spectrum of reasoning skills, spanning both discovery-driven and execution-driven cognition.Furthermore, while LatEval is text-based, PuzzleWorld relies heavily on nuanced multimodal cues. This involves a more complex perceptual and reasoning demand than identifying logical gaps in LatEval’s narrative text.
>
> Our error analysis (Section 5.4) highlights failure modes that relate to both lateral and vertical thinking in PuzzleWorld. For instance, we observe “myopic commitment,” where models fail to explore alternative hypotheses, and weaknesses in multimodal vertical reasoning, such as difficulties in sketching or spatial manipulation. We will incorporate a discussion of LatEval and its relation to PuzzleWorld into the revised draft.
>
> [1] Sun, et al. "Mitigating visual forgetting via take-along visual conditioning for multi-modal long cot reasoning." ACL 2025.

---

> > ### Comment · Reviewer_s7Za · 2025-11-26
> >
> > Thanks for the author's reply. I look forward to seeing the revised paper during the rebuttal process.

---

> > > ### Author Response · Authors · 2025-11-27
> > > **Updated Manuscript & Case Study on Visual Forgetting**
> > >
> > > We thank the reviewer again for their time. We have just uploaded a revised manuscript based on the reviewer's feedback, with the edited sections highlighted in red. We here list the revised sections that are relevant to the reviewer's concerns:
> > > 1. In **Appendix D.2** (Page 16), we include a discussion on whether models' poor performance are due to their limited ability to recognize text in images. In **Appendix E.2** (Page 17), we include the breakdown of GPT-o3's performance by difficulty and performance, with the insight that GPT-o3's performs well on easy, well-structured puzzles, but fail as the diagrams' complexity increase.
> > > 2. In **Section 2** (Page 3), we briefly include a comparison against LatEval in our related work, emphasizing that, while LatEval mainly evaluates lateral thinking in text narratives, PuzzleWorld comprehensively evaluates both lateral and vertical thinking.
> > >
> > > Additionally, in **Appendix D.2, Figure 10** (Page 16), we include a case study on the "visual forgetting" error pattern as observed in GPT-o3's solution with and without the transcribed text. In this puzzle titled "Just Add Half a Scam", GPT-o3 successfully solved the puzzle with the transcribed text clues. However, when relying solely on the images, GPT-o3 correctly decoded the intermediate text clues but failed to extract the final answer, as it could not reliably reference the text within the image -- ultimately hallucinating an incorrect solution.
> > >
> > > Again, we thank the reviewer for their detailed feedback, which has significantly improved our paper's completeness and rigor. Please let us know if there is any further information we could provide :)

---

### Official Review · Reviewer_vb72 · 2025-10-31

**Soundness:** 4
**Presentation:** 3
**Contribution:** 3
**Rating:** 8
**Confidence:** 4

**Summary:**

The authors introduce a dataset of 667 "puzzle hunt"-style puzzles, complete with detailed annotations of reasoning for how to properly solve the puzzle. The authors discuss how this dataset is constructed, and how the annotations are carefully done to ensure no ambiguity in puzzle answers. They evaluate frontier models on the puzzles, and report the performance of both closed- and open-source models. The paper then investigates the effect of fine-tuning on reasoning traces on better performance on other puzzles, and expounds a detailed error analysis of current-generation VLMs.

**Strengths:**

I really like this dataset, both in terms of the intrinsic open-endedness of the puzzles, and the fact that the puzzles have an unambiguous and easy-to-evaluate final answer; I think this is a strong contribution to the field. I also appreciate the attention paid to making the dataset correct and unambiguous. The paper itself is well-written, and the figures are informative. I like the detailed analysis in section 5.4, and appreciate the contamination check in C.1. In all, this is a valuable dataset which was constructed carefully, and an excellent paper analyzing current performance.

**Weaknesses:**

No obvious weaknesses beyond the classic advice that more data points would be better :) Also, it would be nice to re-evaluate on all the new models that came out since the submission deadline, if that's not too hard!

**Questions:**

Would the MIT Mystery Hunt archives be a good source to expand a dataset like this?

---

> ### Author Response · Authors · 2025-11-23
> **Response to Reviewer vb72**
>
> We thank the reviewer for their thoughtful and encouraging feedback, and we are glad that they found PuzzleWorld valuable! We agree that expanding the dataset further would strengthen the benchmark—PuzzledPint releases new puzzles monthly, which would allow us to continually grow PuzzleWorld while also helping mitigate potential data contamination.
>
> **W1. It would be nice to re-evaluate on all the new models that came out since the submission deadline.**
>
> Our current evaluation (Table 2) covers the major frontier multimodal reasoning model families that are available, including GPT-o3, Claude Opus 4, and Grok 4. With the recent release of Gemini 3 Pro, we will additionally benchmark it on PuzzleWorld and will report the results shortly!
>
> **W2. Would the MIT Mystery Hunt archives be a good source to expand a dataset like this?**
>
> Thank you for the excellent suggestion! We did consider the MIT Mystery Hunt during the early design of PuzzleWorld. While it is an attractive source of challenging, high-quality puzzles, several considerations led us not to include it in the initial release:
> - Unlike PuzzledPint (released under Creative Commons), MIT Mystery Hunt puzzles do not have a centralized copyright policy. Permissions would need to be obtained from individual authors, which adds substantial overhead.
> - Mystery Hunt puzzles frequently involve audio, video, interactive web tools, custom code, and even physical components. These are exciting modalities for future generalist AI systems, but they are difficult to standardize under PuzzleWorld’s stepwise annotation format and are not well supported by current frontier models.
> - Mystery Hunt puzzles are intentionally designed to challenge top teams and often require multiple people to solve each puzzle. Given that frontier models already struggle on the entry-level PuzzledPint puzzles, including much harder Mystery Hunt puzzles would likely provide limited practical diagnostic value at this stage.
>
> That said, as models and annotation tools improve, extending PuzzleWorld to incorporate more advanced modalities and higher-difficulty sources—potentially including Mystery Hunt puzzles—would offer a richer evaluation suite. For the present, we prioritize the more accessible PuzzledPint puzzles, as they are both suitably challenging and sufficiently structured to support reliable stepwise annotation and informative intermediate diagnostics.

---

> > ### Author Response · Authors · 2025-11-27
> > **Updated Manuscript & Gemini Benchmarking Results**
> >
> > We thank the reviewer again for their time and their patience as we conduct additional benchmarking on Gemini-3-Pro. We have just uploaded a revised manuscript based on the reviewer's feedback, with the edited sections highlighted in red. We here list the revised sections that are relevant to the reviewer's concerns:
> > 1. In **Table 2** (Page 7), we include additional benchmarking results with Gemini-3-Pro. We observe that the latest version of Gemini outperforms GPT-o3, the current best performing model on PuzzleWorld. On Gemini-3-Pro's modality level performance, we observe similar patterns with other models, achieving highest accuracy on text puzzles and lowest accuracy on visual ones. Nevertheless, Gemini-3-Pro still exhibit significantly lower performance compared to human enthusiasts. We include a table of Gemini's performance below for your convenience.
> > 2. In **Appendix C** (Page 15), we include a discussion on the potential and challenges with extending PuzzleWorld to include MIT Mystery Hunt.
> >
> > | Model | Overall Acc | Overall Step | Text Acc | Text Step | Visual Acc | Visual Step | Structured Acc | Structured Step |
> > | :------ | :---------: | :-----------: | :------: | :---------: | :--------: | :----------: | :------------: | --------------: |
> > | Gemini 3 Pro| 18.00 | 39.99 | 20.30 | 39.34 | 14.71 | 38.81 | 20.25 | 39.99 |
> > | GPT-o3 | 14.22 | 39.81 | 15.16 | 39.92 | 8.96 | 33.38 | 13.53 | 41.28 |
> > | Human Enthusiast | 44.44 | 51.70 | 44.14 | 52.58 | 44.00 | 52.20 | 54.17 | 57.81 |
> >
> > Again, we thank the reviewer for their detailed feedback, which has significantly improved our paper's completeness and rigor. Please let us know if there is any further information we could provide :)

---

### Official Review · Reviewer_NwBy · 2025-11-01

**Soundness:** 3
**Presentation:** 3
**Contribution:** 3
**Rating:** 6
**Confidence:** 4

**Summary:**

This paper presents PUZZLEWORLD, a benchmark of 667 real-world puzzlehunt problems for evaluating multimodal, open-ended, multi-step reasoning. Each puzzle includes detailed reasoning traces, modality and cognitive-skill labels, enabling fine-grained diagnostic analysis. Experiments show that even frontier models achieve only 1–14% final accuracy, while fine-tuning on reasoning traces improves stepwise accuracy from 4.8% to 11%. The benchmark is clear, rigorous, and highly relevant to general reasoning research.

**Strengths:**

(1) Proposes the first large-scale open-ended puzzlehunt benchmark, transforming real Puzzled Pint puzzles into machine-readable tasks that test discovery-driven reasoning.

(2) A two-stage GPT-4o + human verification pipeline ensures 96.5% correctness and no contamination.

(3) Evaluates eight major models with both final and stepwise metrics, identifying systematic weaknesses (myopic reasoning, language bottleneck, lack of sketching).

(4) Reasoning-trace supervision improves intermediate reasoning and transfers to Rebus and MathVista tasks.

**Weaknesses:**

(1) The paper omits discussion of FINEREASON (Chen et al., 2025; arXiv:2502.20238), which also studies reflective puzzle reasoning through step decomposition. Comparing PUZZLEWORLD’s open-ended, multimodal puzzles with FINEREASON’s structured logic ones would clarify its unique scope.

(2) Limited ablations on annotation noise and prompting baselines; fine-tuning details appear only in the appendix.

(3) Dataset excludes audio/video modalities and depends on OCR, limiting breadth.

**Questions:**

– Please add a short discussion contrasting PUZZLEWORLD with FINEREASON, emphasizing that PUZZLEWORLD extends structured reflective reasoning to multimodal, discovery-driven contexts.

– Evaluate robustness to imperfect annotations and potential cross-benchmark transfer.

– Future work could merge PUZZLEWORLD’s multimodal puzzles with reflective reasoning frameworks for richer diagnostics.

---

> ### Author Response · Authors · 2025-11-23
> **Response to Reviewer NwBy (1/2)**
>
> We thank the reviewer for their detailed feedback! We appreciate the reviewer's time in writing the review, and we are happy to respond to the concerns raised.
>
> **W1. Dataset excludes audio/video modalities and depends on OCR, limiting breadth.**
>
> Thank you for raising these important points. We agree that PuzzleWorld currently excludes richer modalities such as audio, video, or interactive environments. These are compelling directions, and we are interested in supporting them in future iterations. However, they present challenges for current PuzzleWorld’s step-wise annotation format, which is considerably harder to standardize for dynamic modalities like video. We prioritize the annotation standardization because it enables measurable intermediate progress and fine-grained reasoning analysis, which is especially important given the difficulty of PuzzleWorld.
>
> We also acknowledge that PuzzleWorld performance can be influenced by a model’s OCR capabilities. Although each puzzle includes a human-transcribed “flavor text”, this may not capture all textual content present. To assess whether OCR is a limiting factor, we selected a 5% subset of strictly text-only puzzles, manually transcribed all textual content, and compared GPT-o3’s performance with and without the full transcription. A paired t-test showed no significant difference ($t = 0.52, p > 0.1$), suggesting that OCR limitations are not a major bottleneck for the frontier models.
>
> That said, we did observe isolated cases where GPT-o3 improved from partial to full solutions once transcription was provided. Upon examination, these instances appear to stem from model hallucination in later stages of reasoning when relying only on images, whereas the transcription helps anchor its reasoning more reliably. This aligns with patterns of visual forgetting observed in prior work such as [1]. We will incorporate this analysis and caveat into the revised draft.
>
> **W2. Please add a short discussion contrasting PUZZLEWORLD with FINEREASON, emphasizing that PUZZLEWORLD extends structured reflective reasoning to multimodal, discovery-driven contexts.**
>
> We thank the reviewer for highlighting this related work. We will add a discussion contrasting PuzzleWorld and FineReason in the revised draft. In brief, the two benchmarks target complementary aspects of stepwise reasoning. FineReason focuses on structured, text-based puzzles with explicit rules, enabling precise state verification and surgical analysis of multi-step logical reasoning. In contrast, PuzzleWorld emphasizes multimodal, discovery-driven reasoning in which the puzzle rules are implicit and must be inferred by the solver. This setting requires a broader range of cognitive skills, including pattern recognition, spatial reasoning, and commonsense interpretation. As a result, PuzzleWorld offers a more holistic evaluation of generalist reasoning capabilities, and we adopt stepwise natural-language annotations with an LLM-judge to approximate intermediate state checks in the absence of explicit rule structures.
>
> [1] Sun, et al. "Mitigating visual forgetting via take-along visual conditioning for multi-modal long cot reasoning." ACL 2025.

---

> > ### Author Response · Authors · 2025-11-23
> > **Response to Reviewer NwBy (2/2)**
> >
> > **W3. Evaluate robustness to imperfect annotations and potential cross-benchmark transfer.**
> >
> > We appreciate the reviewer for raising this concern. We will include an ablation on the effect of annotation noise on our step-level grading accuracy: To simulate annotation noise, we select 10% of PuzzleWorld and use GPT-5-mini to paraphrase each step of the ground truth annotation. Then, we evaluate GPT-o3’s output on both the original and paraphrased ground truth. We observe strong positive correlation between the two sets of scores ($r = 0.898, p < 0.001$) with a mean absolute error of 0.056. This indicates that our evaluation pipeline is robust to moderate noise in the stepwise annotations.
> >
> > Regarding cross-benchmark transfer, we have demonstrated that PuzzleWorld-trained reasoning traces improve performance on both Rebus puzzles and MathVista, showing transfer across distinct multimodal tasks (Table 4). As noted in the draft, a caveat is that finetuning failed to improve performance on MathVista’s “Textbook QA” subset, as these items are out-of-domain relative to the reasoning skills emphasized in PuzzleWorld.
> >
> > **W4. Future work could merge PUZZLEWORLD’s multimodal puzzles with reflective reasoning frameworks for richer diagnostics.**
> >
> > We agree with the reviewer that this is an interesting direction! PuzzleWorld and reflective reasoning frameworks such as FineReason share conceptual similarities in intermediate state checking. A caution to note here is that PuzzleWorld’s open-ended and multimodal puzzles often involve diverse, creative reasoning steps that are difficult to formalize as explicit rules, making it challenging to achieve the same level of rule-based granularity as in more structured puzzle domains in FineReason. That said, integrating reflective reasoning paradigms with PuzzleWorld could yield richer diagnostic tools and potentially benefit training of frontier reasoning models. We see this as an exciting avenue for future work and will highlight it in the revised draft.

---

> > > ### Author Response · Authors · 2025-11-27
> > > **Updated Manuscript**
> > >
> > > We thank the reviewer again for their time. We have just uploaded a revised manuscript based on the reviewer's feedback, with the edited sections highlighted in red. We here list the revised sections that are relevant to the reviewer's concerns:
> > > 1. In **Appendix D.2** (Page 16), we include a discussion on whether OCR capabilities limits frontier multimodal reasoning models' performance on PuzzleWorld, as we have described in our comment above.
> > > 2. In **Section 2** (Page 3), we briefly include a comparison against FineReason in our related work. We further included your suggestion on incorporating reflective reasoning frameworks with PuzzleWorld in our future work section (**Appendix C**, Pages 14-15)
> > > 3. We include in **Appendix F.4** (Page 22) a discussion on the robustness of our stepwise scoring mechanism against human annotation noise.
> > >
> > > Again, we thank the reviewer for their detailed feedback, which has significantly improved our paper's completeness and rigor. Please let us know if there is any further information we could provide :)

---

> > > > ### Comment · Reviewer_NwBy · 2025-11-28
> > > > **Thanks for your response.**
> > > >
> > > > Thanks for your response. Overall, I believe this submission is acceptable for publication, though it does not yet meet the criteria for a rating of 8 IMO.

---

### Official Review · Reviewer_8chG · 2025-11-02

**Soundness:** 2
**Presentation:** 3
**Contribution:** 2
**Rating:** 2
**Confidence:** 4

**Summary:**

The paper introduces PUZZLEWORLD, a benchmark compiles 667 real puzzlehunt problems to test open-ended multimodal reasoning, adding stepwise solution traces, modality and skill tags, and an LLM-judge for intermediate scoring; frontier models perform poorly overall (best ≈14% answer accuracy, ≈40% stepwise), fine-tuning on reasoning steps improves stepwise scores but not final accuracy.

**Strengths:**

- open-ended puzzles obtained from real world, filling a current research gap.

- the paper demonstrated a clear performance gap that stresses present models; best model only has ~14% answer accuracy

**Weaknesses:**

- the work is very interesting, but unfortunately not very thoroughly done, for examples:
    - The “easy/medium/hard” tags come from original puzzle metadata rather than a benchmark-defined rubric. There is no formal difficulty calibration, no solver-time distribution analysis, and no additional validation along the annotation.

   - Humans are grouped into novice/enthusiast/expert tiers, but there is no modality-level breakdown and no step-level scoring symmetry with models. In addition, the human evaluation is only for 5% of the samples, thus the comparison in table 2 is not as informative as it appears.

- The benchmark comes with one gold standard chain per puzzle , yet there is a chance the puzzlehunts can have multiple legitimate solve paths. There is no mechanism for crediting alternative correct partial chains or heuristic leaps, thus the evaluation of the reasoning paths can be quite limited.

- the significance of the benchmark may be limited in the sense that the problems and solutions are quite scoped with the culture of puzzlehunt. These specialized cognitive challenges might not well generalize to a broader scope.

**Questions:**

I will recommend the authors to increase the rigor of the benchmark, as discussed above.

---

> ### Author Response · Authors · 2025-11-23
> **Response to Reviewer 8chG (1/2)**
>
> We thank the reviewer for their detailed feedback! We appreciate the reviewer's time in writing the review, and we are happy to respond to the concerns raised.
>
> **W1. The “easy/medium/hard” tags come from original puzzle metadata rather than a benchmark-defined rubric.**
>
> Thank you for raising this point. We agree that difficulty calibration is an important aspect of benchmark design. Our use of the original PuzzledPint difficulty labels is a deliberate choice. According to the PuzzledPint website, all submitted puzzles undergo internal playtesting by an editorial team, where the difficulty tags are revised and finalized by expert editors and reflect a community-standard difficulty assessment. We thus treat them as expert annotations rather than ad-hoc metadata. From the AI model’s perspective, the labels also meaningfully reflect task difficulty, since GPT-o3’s accuracy decreases as difficulty level increases:
> | Difficulty | Accuracy    | Stepwise
> |:------------------- |:--------------------------:|:--------------------------:|
> | Easy | 18.46% | 40.58% |
> | Medium | 14.76% | 39.38% |
> | Hard | 9.64% | 39.71 % |
>
> A valid concern is that human-perceived difficulty and AI-perceived difficulty might not align. For example, certain diagram-heavy puzzles may be straightforward for humans but require nontrivial spatial reasoning for AIs. To investigate this, we break down GPT-o3’s performance by modality and difficulty:
>
> | Difficulty           | Text    | Visual | Structured |
> |:---------------- |:--------------------:|:--------------------:|:--------------------:|
> | Easy | 19.5% | 11.9% | 20.4% |
> | Medium | 15.3% | 8.9% | 13.5% |
> | Hard| 10.6% | 7.6% | 9.6% |
>
> Within each modality, accuracy consistently decreases with difficulty. Notably, GPT-o3 performs relatively well on easy structured puzzles – likely due to their clean, regular layouts – but its performance drops sharply on harder structured puzzles where the diagrams become more irregular and visually complex. Overall, both aggregated and modality-level results show that PuzzledPint’s difficulty labels provide a meaningful and consistent difficulty metric for PuzzleWorld. We will include this discussion in the final draft of this paper.
>
> **W2. There is no modality-level breakdown and no step-level scoring symmetry with models for human baselines.**
>
> Thank you for the helpful suggestion! We will update the paper with a modality-level breakdown of the human baseline, as provided below. As in the paper, we assume experts have 100% accuracy. We observe that, unlike AI models, visual and structured modalities do not pose additional difficulty for human solvers, further highlighting the multimodal limitations of current foundation models.
>
> |Experience Level | Text (Acc) | Text (Step) | Visual (Acc) | Visual (Step) | Structured (Acc) | Structured (Step) |
> |:---------------- |:--------------------:|:--------------------:|:--------------------:|:--------------------:|:--------------------:|:--------------------:|
> | Novice | 16.98% | 25.32% | 11.00% | 22.70% | 16.67% | 24.92% |
> | Experienced | 44.14% | 52.58% | 44.00% | 52.20% | 54.17% | 57.81% |
> |Expert| 100.0% | 100.0% | 100.0% | 100.0% | 100.0% | 100.0% |
>
> After revisiting the human baseline results, we found that each modality has more than 40 human attempts (since we assign multiple participants to each puzzle to ensure consistency), which we consider sufficient to expose modality-level human-AI performance gaps. Regarding step-level scoring, the same LLM-judge protocol is applied to both human and model solutions, ensuring that the comparison is fair and step-symmetric. We will make this more explicit in the revision.
>
> We also acknowledge that the human baseline covers only a subset of the full benchmark. Given our resource constraints, we prioritized depth and quality in the human evaluation rather than exhaustiveness. To reduce individual variability in a task as open-ended as PuzzleWorld, each sampled puzzle was solved by at least two independent participants, each allotted up to one hour and asked to produce detailed reasoning traces. Despite the limited sample size, the resulting baselines reveal clear separations between human tiers and current models—for example, a >30% gap between human enthusiasts and GPT-o3. This gap is substantially larger than the uncertainty introduced by the sample size, suggesting that the current human results provide a reliable directional reference point. We will clarify both the strengths and limitations of the current human study in the revised paper.

---

> > ### Author Response · Authors · 2025-11-23
> > **Response to Reviewer 8chG (2/2)**
> >
> > **W3. There is no mechanism for crediting alternative correct partial chains or heuristic leaps, thus the evaluation of the reasoning paths can be quite limited.**
> >
> > In puzzlehunts, the solution steps are intentionally designed to be interlocking and sequentially dependent. It is thus hard to reach the correct final answer while following a completely different solution path, and correct partial chains must partially converge with the annotated ground-truth steps, even if the solver temporarily deviates or makes logic leaps.
> >
> > That said, human puzzlehunters do commonly make educated guesses that skip over intermediate steps. For example, after identifying three of the four final answer letters (“I _ L R”), a solver might correctly infer “ICLR” without fully completing the preceding clue. Our step-level scoring explicitly accounts for this behavior: we identify the latest ground-truth step that appears in the candidate solution. In the above scenario, because the solver correctly arrives at the final answer, our scoring grants full credit, even if several intermediate steps were omitted or approximated.
> >
> > As a proxy for evaluating how robust our step-level crediting is to alternative partial chains, we randomly sampled 10% of PuzzleWorld and automatically paraphrased GPT-o3’s solutions using GPT-5-mini. We then compared the LLM-judge’s step-level scores on the original and paraphrased solutions. We observe strong positive correlation between the two sets of scores ($r = 0.885$, $p < 0.001$) with a mean absolute error of 0.066, indicating a high degree of score stability across semantically equivalent but structurally varied solution traces. Although this does not exhaustively cover all possible reasoning paths, it indicates that the judge can reliably credit alternative solution trajectories. We will include the above discussion in the revised paper.
> >
> > **W4. The significance of the benchmark may be limited in the sense that the problems and solutions are quite scoped with the culture of puzzlehunt.**
> >
> > We appreciate the reviewer’s concern and agree that PuzzleWorld is not drawn from real-world tasks such as mathematics or physics. Our motivation for using puzzlehunts is precisely their abstraction, which is desirable for assessing generalist, open-ended reasoning, and reducing pattern memorization. For example, prior work has shown that symbolic variants of math datasets can reveal brittle reasoning behavior in models that otherwise appear strong [1]. PuzzleWorld’s  abstraction thus helps isolate and measure a model’s generalist and genuine reasoning ability beyond domain-specific patterns.
> >
> > Existing reasoning benchmarks–ranging from math and coding tasks--typically operate in closed-ended, well-defined environments. These datasets assess correctness under fixed rules, but they do not require models to navigate underspecified or open-ended problem spaces. In contrast, many real-world tasks (e.g., exploratory data analysis, investigative research, science discovery) involve ambiguous signals, multiple possible solutions, and iterative hypothesis testing.
> >
> > PuzzleWorld fills this gap in two ways that previous abstract reasoning benchmarks don’t. First, unlike traditional puzzle datasets (e.g. Sudoku) that evaluate within a limited rule set, our puzzlehunt puzzles require leveraging diverse reasoning competencies (Figure 2) to dynamically form and evaluate hypotheses from nuanced multimodal signals. Second, our manually annotated reasoning traces enable systematic analysis of model behavior in these open-ended settings, revealing characteristic failure modes of frontier models such as “myopic commitment” (Section 5.4). Our transfer experiments (Table 4) has also shown that capabilities exercised in PuzzleWorld also transfer to real-world benchmarks, suggesting that puzzlehunt-style reasoning is a useful proxy for generalist reasoning. Nonetheless, we acknowledge that no single benchmark can represent all forms of reasoning, and we will highlight this scope limitation more explicitly.
> >
> > [1] Mirzadeh et al. "GSM-Symbolic." ICLR 2025.

---

> > > ### Author Response · Authors · 2025-11-27
> > > **Updated Manuscript**
> > >
> > > We thank the reviewer again for their time. We have just uploaded a revised manuscript based on the reviewer's feedback, with the edited sections highlighted in red. We here list the revised sections that are relevant to the reviewer's concerns:
> > > 1. In **Appendix E.2** (Page 17), we revised the manuscript to include a discussion on the alignment between PuzzledPint's difficulty labels with models' performance on PuzzleWorld, as described in our comment above.
> > > 2. In **Table 2** (Page 7), we include the modality-level breakdown for human baseline. We additionally include a discussion on the strengths and limitations of our human baseline in **Appendix F.3** (Page 22).
> > > 3. In **Appendix D.3** (Page 16), we include a discussion on how the stepwise scoring mechanism handles alternative partial solutions.
> > > 4. In **Appendix D.1** (Page 15), we include a discussion on the significance of abstraction in PuzzleWorld, compared against close-ended datasets such as math and physics and other puzzle benchmarks.
> > >
> > > Again, we thank the reviewer for their detailed feedback, which has significantly improved our paper's completeness and rigor. Please let us know if there is any further information we could provide :)

---

### Comment · Area_Chair_MUsw · 2025-11-27
**Remarks from AC**

**To Reviewers:**

Thank you for providing the initial reviews. If you haven't done so, please read the authors' responses and engage in the discussion. If the authors have addressed your concerns, please let them know. Otherwise, please state your remaining concerns.

**To Authors:**

One of the reviewers has already replied to your rebuttal and is waiting for the updated manuscript. Could you please upload it before the discussion period ends?

Best,

AC

---

> ### Author Response · Authors · 2025-11-27
> **Thank you for the reminder!**
>
> We thank the AC for their kind reminder! We have just uploaded a revised manuscript based on reviewers' feedback, with the edited sections highlighted in red. We will leave additional comments to each reviewer to help them identify sections relevant to their suggestions.

---

### Author Response · Authors · 2025-12-03
**Authors Final Remarks**

We thank the reviewers for their constructive feedback on PuzzleWorld. We acknowledge that, due to the recent identity leakage, we cannot engage in further direct discussion with the reviewers. Therefore, we provide the summary below on how we have incorporated the reviewers' suggestions into the final manuscript.

-----

**Validation of Difficulty Labels (Reviewer 8chG):**

We address the concern that PuzzleWorld's difficulty labels are derived from data sources rather than a benchmark-defined rubric. In **Appendix E.2 (Page 17)**, we discuss how these labels are treated as expert annotations and demonstrate that the benchmarked models’ performance aligns with the labeled difficulty.

**Human Baseline Modality-Level Breakdown (Reviewer 8chG):**

We have updated **Table 2 (Page 7)** to include a modality-level breakdown for the human baseline, and in **Appendix F.3 (Page 22)** we discusses the strengths and limitations of our human baseline.

**Handling Alternative Solution Paths in Stepwise Scoring (Reviewer 8chG):**

To address concerns regarding alternative partial chains or heuristic leaps, we have added a discussion in **Appendix D.3 (Page 16)** detailing how our stepwise scoring mechanism successfully accounts for valid alternative partial solutions.

**PuzzleWorld's Significance Beyond Puzzlehunts (Review 8chG):**

Regarding the concern that PuzzleWorld might be limited to the culture of puzzlehunts, we include in **Appendix D.1 (Page 15)** a discussion on why PuzzleWorld's domain-agnostic abstraction and open-ended nature is desirable and significant compared to closed-ended benchmarks in math or physics.

**PuzzleWorld's OCR Dependencies (Reviewers NwBy, s7Za):**

In **Appendix D.2 (Page 16)**, we discuss that while OCR is not the primary bottleneck for frontier reasoning models on PuzzleWorld, it can lead to occasional "visual forgetting." We have included a case study illustrating this error pattern in **Figure 10 (Page 16)**.

**Robustness to Annotation Noise (Reviewer NwBy):**

We have added a discussion in **Appendix F.4 (Page 22)** demonstrating how our stepwise scoring mechanism remains robust in the presence of potential imperfect human annotations.

**Additional Benchmarking (Reviewer vb72):**

We have updated **Table 2 (Page 7)** to include results for Gemini-3-Pro. Notably, this model has surpassed GPT-o3, establishing a new state-of-the-art performance on PuzzleWorld.

**Related Works (Reviewers NwBy, s7Za):**

Per the reviewers’ suggestions, we have updated **Section 2 (Page 3)** to include a comparison of PuzzleWorld against FineReason and LatEval.

**Future Directions (Reviewers NwBy, vb72):**

In **Appendix C (Pages 14-15)**, we have expanded our discussion on the potential and challenges of future directions, including extending PuzzleWorld with reflective reasoning frameworks for diagnostics and the inclusion of more complex challenges like the MIT Mystery Hunt.

-----
We acknowledge the exceptional circumstances of this year's review cycle. We again thank all the reviewers for their time and constructive feedback on PuzzleWorld, and we hope this summary assists the AC in the final decision-making process.

---

### Meta-Review · Area_Chair_Caph · 2026-01-04

**Summary:**

Reviewers generally agree that the presented benchmark, PuzzleWorld, which takes inspiration from puzzlehunts and involves solving puzzles with implicit rules, is a compelling addition to the family of generalist AI benchmarks. Concerns raised during the review period included:

### Difficulty assessment and general benchmarking of puzzles
* Reviewer 8chG was concerned about how the difficulty of puzzles was being assessed, as well as understanding the human baseline in more detail. The authors convincingly showed that they could use alternative ways of scoring difficulty, which all track with the metadata that they originally used for difficulty generation. They similarly provided a modality-level breakdown of the human performance in the appendix.

### Multimodal integrations and robustness
* Multiple reviewers were concerned about OCR (visual) dependencies in the puzzleworld benchmark. The authors showed that OCR is an issue, but not the main bottleneck for performance in the benchmark. They similarly showed the vision dependency for MLLMs is an issue, but not the main bottleneck for solving the puzzles.

### Related works
* Multiple reviewers brought up related pieces of work that should be discussed in an updated manuscript. The authors included these in the manuscript, and successfully explained the differences between PuzzleWorld and these other benchmarks.

Overall, this seems like a compelling new benchmark and I recommend accepting this work.

**Reviewer Concerns:**

Addressed by the rebuttal:
* Assessments of difficulty for the puzzles.
* Related work contextualization.
* Future work and limitations section.
* Analysis of failure modes related to OCR.

Still outstanding
* relevance of puzzleworld to other tasks. It is not clear if solving PuzzleWorld will mean that agents are significantly more general in a way that allows them to solve other tasks. However, that criticism can be applied to most benchmarks that are aiming to measure "general intelligence", so I do not believe it is a major barrier.
* there is no obvious way to extend this into an interactive environment, or one which integrates video or audio.

**Reviewer Scores:**

I expect that every reviewer would have increased or maintained their score during the rebuttal period. Reviewer 8chG, who was the most negative, also had their main points convincingly addressed by the authors (who included a bunch of extra analyses speaking to the reviewer's concerns).

Given that only 8chG initially voted to reject the paper (with all others voting to accept), I am therefore inclined to accept the paper.

---

### Decision · Program_Chairs · 2026-01-26

Accept (Poster)